# Tgfβ signaling is required for tenocyte recruitment and functional neonatal tendon regeneration

Deepak A Kaji, Kristen L Howell, Zerina Balic, Dirk Hubmacher, Alice H Huang*

Department of Orthopaedics, Icahn School of Medicine at Mount Sinai, New York, United States

**Abstract** Tendon injuries are common with poor healing potential. The paucity of therapies for tendon injuries is due to our limited understanding of the cells and molecular pathways that drive tendon regeneration. Using a mouse model of neonatal tendon regeneration, we identified TGFβ signaling as a major molecular pathway that drives neonatal tendon regeneration. Through targeted gene deletion, small molecule inhibition, and lineage tracing, we elucidated TGFβ-dependent and TGFβ-independent mechanisms underlying tendon regeneration. Importantly, functional recovery depended on canonical TGFβ signaling and loss of function is due to impaired tenogenic cell recruitment from both *Scleraxis*-lineage and non-*Scleraxis*-lineage sources. We show that TGFβ signaling is directly required in neonatal tenocytes for recruitment and that TGFβ ligand is positively regulated in tendons. Collectively, these results show a functional role for canonical TGFβ signaling in tendon regeneration and offer new insights toward the divergent cellular activities that distinguish regenerative vs fibrotic healing.

## Introduction

Tendons connect muscle to bone and function to transmit muscle forces to the skeleton. Tendon function is enabled by a specialized extracellular matrix predominantly composed of highly aligned type I collagen fibrils (*Voleti et al., 2012*). Although healthy tendon can normally resist high mechanical loads, mechanical properties are permanently impaired after injury due to its minimally regenerative potential (*Voleti et al., 2012*). This loss of function can lead to chronic pain, decreased quality of life, and increased risk of re-rupture. Current treatment options remain limited and there are almost no cell or biological treatments to improve tendon repair or induce regeneration.

To date, the majority of injury models to study tendon healing result in scar-mediated healing since adult tendon does not regenerate (*Ackerman et al., 2019*; *Dyment et al., 2014*; *Dyment et al., 2013*; *Howell et al., 2017*; *Katzel et al., 2011*; *Kim et al., 2011*; *Mass and Tuel, 1991*). Although a few groups showed successful tendon regeneration in model systems such as fetal sheep and MRL/MpJ mice, genetic manipulation is relatively challenging in these systems (*Beredjiklian et al., 2003*; *Paredes et al., 2018*). To overcome these limitations, we previously established a model of tendon regeneration in neonatal mice, that can be directly compared to fibrotic tendon healing in adult mice within the same genetic background (*Howell et al., 2017*). Using lineage tracing, we found that neonatal tendon regeneration is driven by tenocyte proliferation, recruitment, and differentiation leading to full functional restoration. In contrast, adult tendon healing is defined by the persistence of aSMA+ cells, absence of tenocyte proliferation or recruitment, abnormal differentiation into cartilaginous cells, and loss of functional tendon properties. Having identified cellular processes distinguishing neonatal and adult tendon healing, we now focus on the molecular pathways that regulate neonatal tendon regeneration.

*For correspondence:
alice.huang@mssm.edu

**Competing interests:** The authors declare that no competing interests exist.

Although FGF signaling was first established in chick tendon development (*Brent et al., 2003*), the TGFβ pathway subsequently emerged as the most important signaling pathway identified for mammalian tendon formation (*Havis et al., 2016*; *Kuo et al., 2008*; *Pryce et al., 2009*). Members of the TGFβ superfamily of growth factors signal through type II receptors, which then dimerize with type I receptors. The type I receptor then phosphorylates intracellular Smad transcription factors that complex with the co-Smad, SMAD4, to change transcriptional programs (*Shi and Massagué, 2003*). In mouse embryos, TGFβ ligands are expressed by tendon cells and genetic deletion of the TGFβ type II receptor (TβR2) or the TGFβ ligands result in a total absence of tendons (*Havis et al., 2016*; *Kuo et al., 2008*; *Pryce et al., 2009*). TGFβs also induce expression of the tendon transcription factor, *Scleraxis* (*Scx*), in different experimental systems including embryonic limb explants, mesenchymal stem cells, and tendon-derived cells (*Brown et al., 2015*; *Maeda et al., 2011*; *Pryce et al., 2009*).

In addition to its essential role in tendon development and tendon cell differentiation, TGFβ is also a known inducer of fibrotic scar formation in diverse tissues, including adult tendon (*Katzel et al., 2011*; *Kim et al., 2011*; *Thomopoulos et al., 2015*). TGFβ is well established as a driver of myofibroblast differentiation (*Border and Noble, 1994*; *Desmoulière et al., 1993*) and excessive release of TGFβ ligand after injury can also induce tenocyte death (*Maeda et al., 2011*). Given these contradictory roles of TGFβ signaling in both tendon differentiation and scar formation, it is unclear whether TGFβ signaling enacts a positive or negative response in the context of tendon regeneration. To determine the requirement for TGFβ signaling in neonatal tendon regeneration, we used pharmacological TGFβ inhibition and genetic deletion experiments. We identified a role for canonical TGFβ signaling in promoting functional regeneration of neonatal tendon and recruitment of tenogenic cells derived from both *Scx* and non-*Scx* lineages.

## Results

### TGFβ signaling is activated after neonatal injury

To determine whether TGFβ signaling is activated after neonatal injury, we measured gene expression of the TGFβ type II receptor (*Tgfbr2*) and the isoforms *Tgfb1*, *Tgfb2*, and *Tgfb3* at day (d) 3, d7, d14, and d28 post-injury by qPCR. *Tgfbr2* was expressed at all time points after injury and upregulated in injured tendons compared to uninjured controls at d7 and d28 ($p < 0.01$) (*Figure 1A*). *Tgfb1* and *Tgfb2* expression levels were transiently upregulated at alternating time points; however, overall expression levels were quite low for *Tgfb2* expression (several fold lower than *Tgfb1* and *Tgfb3*), suggesting a relatively minor role in regeneration for this ligand. By contrast, *Tgfb3* expression was increased at all time points relative to control (*Figure 1A*). Consistent with the gene expression analysis, western blot analysis showed Smad2/3 phosphorylation (pSmad2/3) in injured tendons at d14, suggesting activation of canonical TGFβ signaling (*Figure 1B*). Analysis of active and total TGFB1 ligand by ELISA showed no change in the amount of active TGFB1 with injury, but a significant increase in total TGFB1 (*Figure 1C*). Collectively, these results suggest activation of TGFβ signaling in neonatal tendon regeneration.

### TGFβ signaling is required for functional regeneration

To test the requirement for TGFβ signaling in functional neonatal tendon regeneration (*Figure 2*), we inhibited TGFβ signaling for 14 days after injury using the well-established small molecule inhibitor SB-431542, which targets the TGFβ type I receptors ALK 4/5/7 (*Araújo-Jorge et al., 2012*; *Callahan et al., 2002*; *Inman et al., 2002*; *Lemos et al., 2015*; *Mercado-Gómez et al., 2014*; *Mohamed et al., 2017*; *Pulli et al., 2015*; *Sato et al., 2015*; *Shi et al., 2017*). Analysis of pSmad2/3 by western blotting showed a significant decrease with SB-431542 treatment when normalized to contralateral control tendons (*Figure 2—figure supplement 1*). In contrast, phospho-p38, which is a Smad-independent mediator of TGFβ signaling was not affected (*Figure 2—figure supplement 1*). Neonatal mice treated with the SB-431542 showed no adverse effects on growth compared to carrier-treated mice and tendons appeared grossly normal (*Figure 2—figure supplement 2*).

To determine the role of TGFβ signaling in functional healing, we first analyzed the gait parameters % brake and % propel, which are highly associated with Achilles tendon function (*Howell et al., 2017*). Carrier-treated mice fully recovered % brake and % propel by d14, consistent with functional

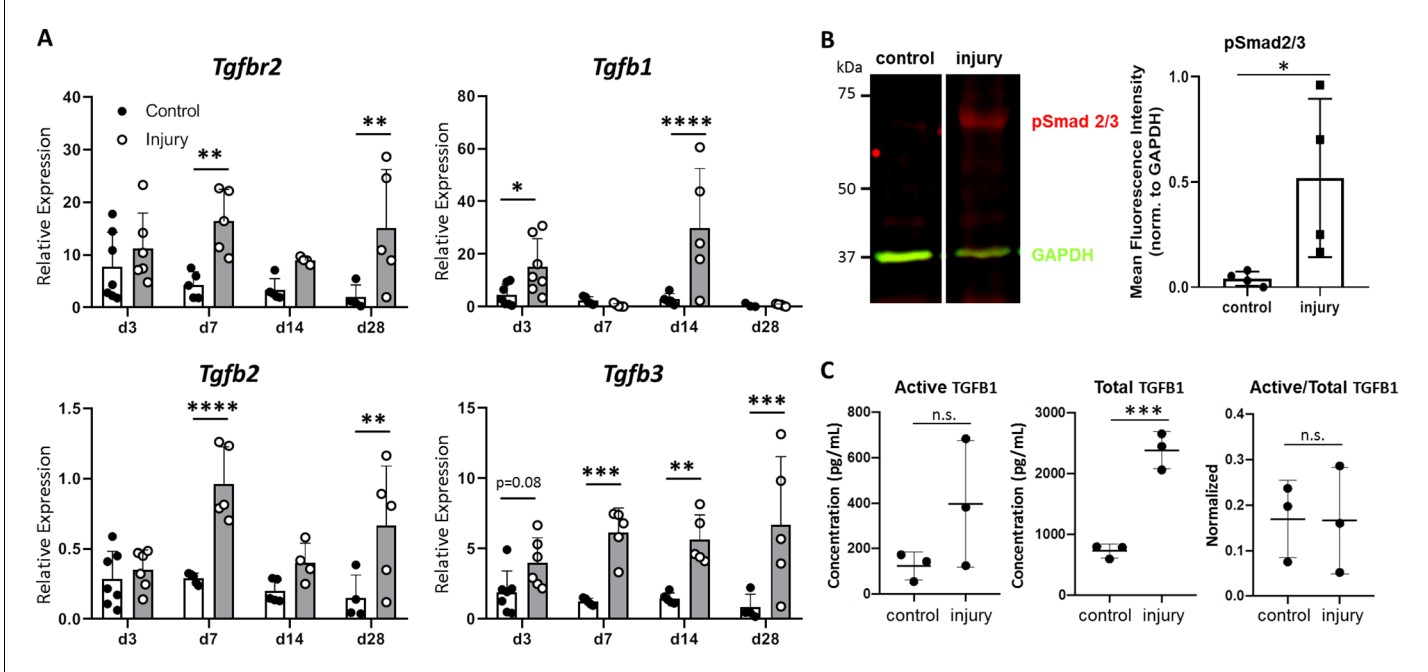

**Figure 1.** TGFβ signaling is activated after neonatal tendon injury. (**A**) Gene expression in control and injured tendons at d3, d7, d14, and d28 post-injury by qPCR showed upregulation of *Tgfbr2* receptor and *Tgfb1*, *Tgfb2*, *Tgfb3* ligands. Expression levels were normalized to *Gapdh* using a standard curve method (n = 5–7 mice). (**B**) Western blot of control and injured tendons at d14 showed Smad2/3 phosphorylation after injury indicative of active TGFβ signaling (3 tendons per sample, n = 3 samples). (**C**) ELISA detection of active and total TGFB1 protein at d14 shows no change in active TGFB1 and increased total TGFB1 with injury (3 tendons per sample, n = 3 samples). *p<0.05, **p<0.01, ***p<0.001, ****p<0.0001.

regeneration (*Figure 2A*). By contrast, both % brake and % propel were impaired relative to the contralateral control limb with SB-431542 treatment. We also observed a significant decrease in %

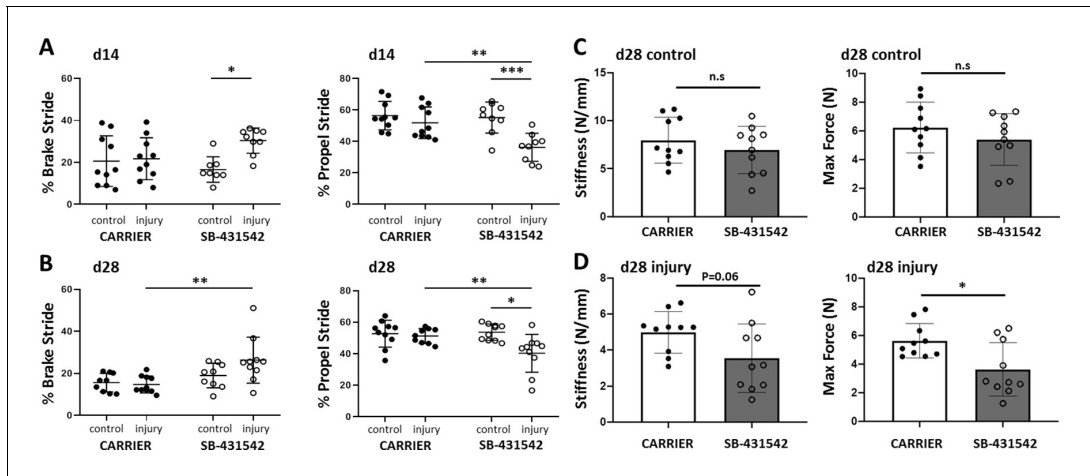

**Figure 2.** TGFβ signaling is required for functional recovery. Gait analysis at (**A**) d14 and (**B**) d28 showed impaired % brake stride and % propel stride after injury with SB-431542 treatment. (**C, D**) Tensile testing at d28 revealed reduced stiffness and max force with SB-431542 treatment. *p<0.05, **p<0.01, ***p<0.001 (n = 8–10 mice).

The online version of this article includes the following figure supplement(s) for figure 2:

**Figure supplement 1.** SB-431542 inhibition modulates canonical TGFβ signaling.

**Figure supplement 2.** Postnatal growth is not affected by SB-431542 treatment.

propel stride relative to the injured limb of carrier-treated animals. Defects in whole limb gait persisted until d28 for both parameters despite cessation of inhibitor treatment at d14 (*Figure 2B*).

To determine the mechanical properties of the healing tissue directly, we performed tensile testing of the tendons at d28. Although mechanical properties in uninjured control tendons were not significantly different with SB-431542 treatment, we observed a reduction in stiffness and max force of injured SB-431542-treated tendons relative to carrier (*Figure 2C and D*). Taken together, these data showed that TGFβ signaling in the first 14 days after injury is required for functional tendon regeneration.

## TGFβ signaling in neonatal tenocytes is required for cell recruitment after injury

We previously found that *Scx*-lineage (Scx[lin]) tenocyte proliferation, recruitment, and differentiation are unique features of the neonatal regenerative response that are not observed during adult healing (*Howell et al., 2017*). To determine the cellular basis for the functional deficits observed with TGFβ inhibition, we next assessed whether tenocyte recruitment is affected by TGFβ inhibition using the *Scx-CreERT2* mouse (*Figure 3*). Differentiated *Scx*-expressing tenocytes were labeled by tamoxifen prior to injury and traced during regeneration with SB-431542 inhibition. Analysis of non-tamoxifen treated *Scx-CreERT2*; ROSA-Ai14 mice affirmed the absence of leakiness by the Cre-allele since TdTomato+ cells were not observed even at 1 year of age (*Figure 3—figure supplement 1*). In carrier-treated, tamoxifen-injected mice, whole mount imaging of hind limbs showed Scx[lin] cells (TdTomato+) occupying the gap space between the original Achilles tendon stubs at d14 after

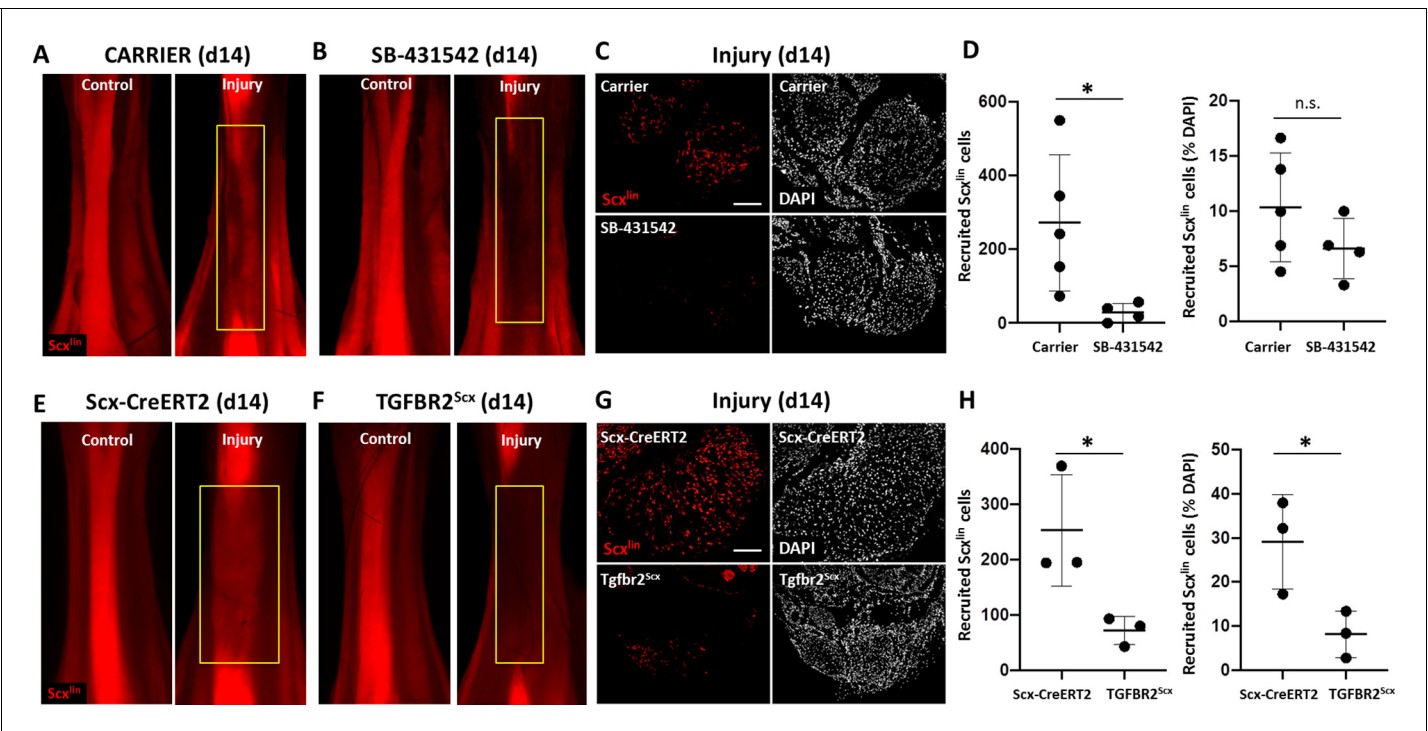

**Figure 3.** TGFβ signaling is required for tenocyte recruitment at d14. (A, B) Whole mount images of control and injured limbs in carrier-treated and SB-431542-treated *Scx-CreERT2*; ROSA-Ai14 mice at d14. (C) Transverse sections through the neotendon (yellow boxes) near the mid-substance and (D) quantification showed reduced Scx[lin], TdTomato+ cell recruitment with SB-431542 treatment, but no difference when normalized to total cells (n = 4–5 mice). (E, F) Whole mount images of control and injured limbs in wild type *Scx-CreERT2*; ROSA-Ai14 and TGFBR2[Scx]; ROSA-Ai14 mice. (G) Transverse sections through the neotendon near the mid-substance and (H) quantification showed reduced Scx[lin], TdTomato+ cell recruitment in TGFBR2[Scx] mutants (n = 3 mice). *p<0.05, n.s. indicates p>0.05. Scale bars: 100 μm.

The online version of this article includes the following figure supplement(s) for figure 3:

**Figure supplement 1.** *Scx-CreERT2* does not label any cells in the absence of tamoxifen.
**Figure supplement 2.** Reduced immunostaining of TGFBR2 in Scx[lin] cells with TGFBR2[Scx] deletion.

transection (*Figure 3A*), while little TdTomato signal was detected in SB-431542-treated limbs (*Figure 3B*). Quantification of transverse sections taken from the midsubstance regions confirmed reduced Scx^lin tenocyte numbers with TGFβ signaling inhibition (*Figure 3C and D*). Interestingly, when Scx^lin tenocytes were normalized to total DAPI-positive cells, this difference was abolished. Quantification of total number of cells confirmed decreased cell number with SB-431542 treatment, suggesting a cell recruitment or cell proliferation defect (1119 ± 252 carrier vs 637 ± 170 SB-431542, p=0.01).

Since SB-431542 treatment targets all cells and we observed reduced total cell number in the neotendon with TGFβ signaling inhibition, we next tested whether neonatal tenocytes directly required TGFβ signaling for their recruitment. We therefore deleted *Tgfbr2* using *Scx-CreERT2* prior to injury and labeled mutant cells by TdTomato (*Tgfbr2^f/f*; *Scx-CreERT2*; ROSA-Ai14; referred to here as TGFBR2^Scx). Since TGFβ signaling is mediated by a single type II receptor (TGFBR2), all TGFβ signaling is abolished with deletion of this receptor. Antibody staining against TGFBR2 confirmed deletion of the receptor in the majority of Scx^lin cells in TGFBR2^Scx injured tendon stubs at d14 (*Figure 3—figure supplement 2*). Consistent with our inhibitor studies, few Scx^lin tenocytes were detected in the neotendon of TGFBR2^Scx mutant tendons compared to *Scx-CreERT2* wild type tendons at d14 post-injury (*Figure 3F–H*). This difference was maintained when Scx^lin cells were normalized to total DAPI-positive cells (*Figure 3H*), since quantification of total cell number showed no difference between wild type and mutant tendon (928 ± 359 WT vs 976 ± 484 TGFBR2^Scx, p=0.9). These data suggest that TGFβ signaling is required for tenocyte recruitment after neonatal injury, rather than it being a consequence of TGFβ inhibition on other cells. Our data further show that while wild type, non-Scx^lin cells are able to compensate for the absence of TGFBR2^Scx knockout Scx^lin cells, this compensation does not occur when TGFβ signaling is systemically inhibited in all cells.

## TGFβ signaling is required for tenocyte migration but not proliferation

We hypothesized that the absence of Scx^lin cell recruitment at d14 with TGFβ inhibition may be due to a defect in cell proliferation at an earlier time point. In a previous study, we showed intense Scx^lin tenocyte proliferation that was localized at the cut site of tendon stubs at d3. To test this hypothesis, we collected *Scx-CreERT2*-labeled limbs at d3 post-injury with SB-431542 treatment as well as TGFBR2^Scx deletion. Consistent with previous findings, transverse sections through the midsubstance gap space confirmed that Scx^lin cells were not detectable at d3 after injury for any condition (not shown). EdU staining of proliferating cells at the cut site of the tendon stubs showed comparable numbers of proliferating Scx^lin tenocytes between carrier-treated and SB-431542-treated mice after injury (*Figure 4A–C*). Similarly, no differences were detected between injured, wild type and TGFBR2^Scx mutants (*Figure 4D–F*). However, total cell proliferation (Scx^lin and non-Scx^lin) was decreased with SB-431542-treatment (p=0.07), while no difference in total cell proliferation was detected in TGFBR2^Scx mutants (*Figure 4C and F*). At this time point, tenocyte proliferation in uninjured control Achilles tendons was almost undetectable (0–1 EdU+/Scx^lin+ cell per section) and was unaffected by SB-431542 treatment or TGFBR2^Scx deletion (*Figure 4—figure supplement 1*). These data suggest that reduced cell number at d14 with inhibitor treatment may be due in part to reduced proliferation of non-Scx^lin cells.

Since proliferation of Scx^lin cells was not affected at d3, we next determined whether TGFβ signaling may be required for tenocyte migration. In vitro scratch assays in tenocyte monolayers were used to induce cell migration (*Figure 5A*). Tenocytes in DMEM alone did not migrate at any time point. Differences in wound closure were not observed between DMEM and DMEM+FBS until 12 hr. In contrast, addition of TGFB1 significantly enhanced cell migration, and differences in wound closure were detected as early as 4 hr after scratching with nearly full closure achieved by 8 hr (*Figure 5A*). To confirm that accelerated migration occurred in Scx^lin cells, we labeled tendons of *Scx-CreERT2* mice at P2 and P3, and isolated cells at P7 from non-injured tendons. We observed enhanced Scx^lin cell migration with TGFB1 treatment (*Figure 5B*). Quantification of Scx^lin proliferating cells detected by EdU showed minimal proliferation with no differences detected across treatment groups (*Figure 5B*). EdU quantification for non-Scx^lin cells also showed no difference (2.89 ± 1.24% DMEM vs 3.46 ± 2.01% DMEM+FBS vs 3.26 ± 1.23% DMEM+FBS+TGFB1, p=0.8).

Collectively, these results suggest that TGFβ signaling is required for recruitment of neonatal tenocytes after injury, and that tenocyte recruitment requires active cell migration rather than

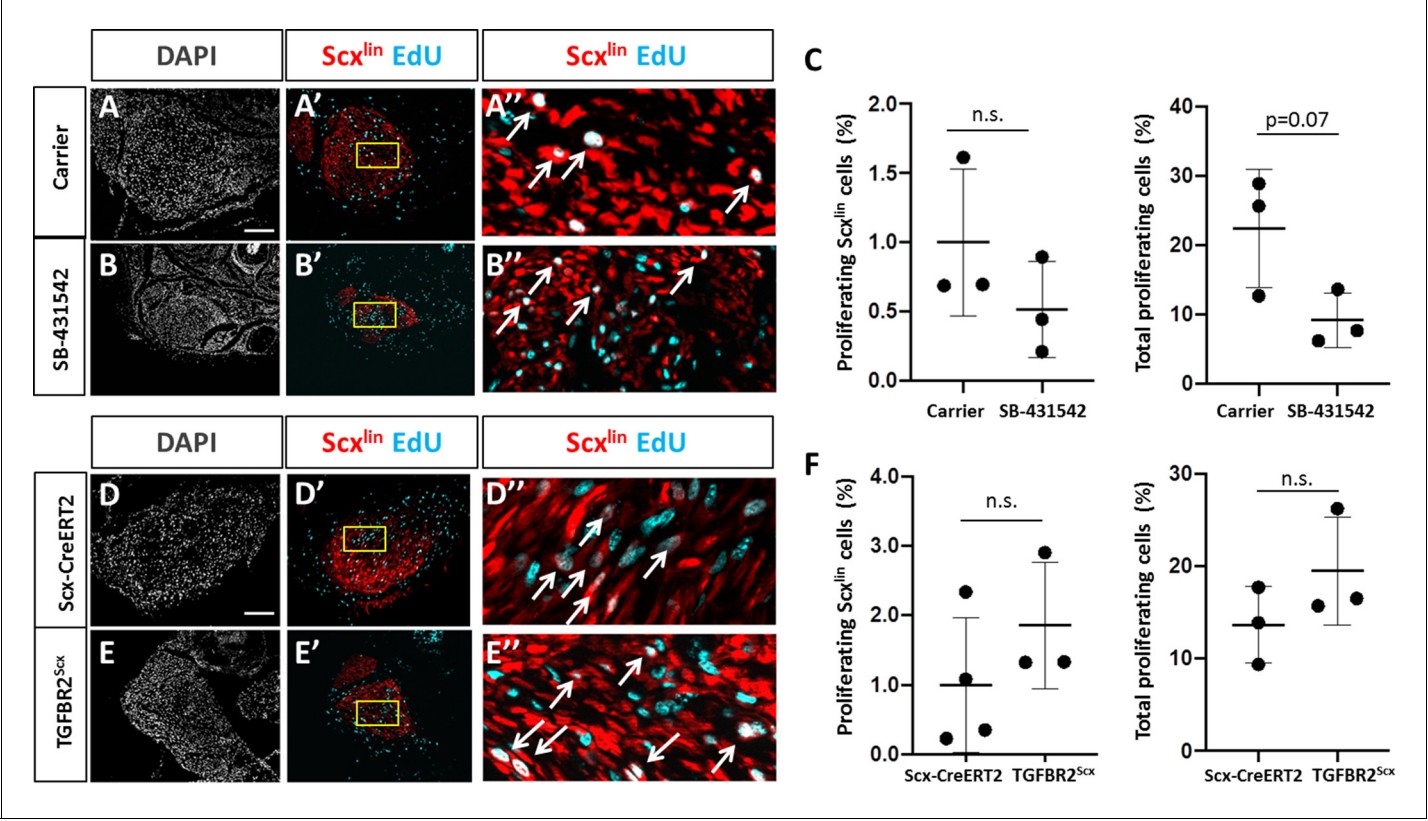

**Figure 4.** TGFβ signaling is not required for tenocyte proliferation. Transverse sections through the cut site of (**A, A', A'**) carrier-treated injured tendon or (**B, B', B'**) SB-431542-treated injured tendon stained for EdU and counterstained with DAPI. (**C**) Quantification of EdU and Scx[lin] overlays showed no difference in Scx[lin] cell proliferation after injury with TGFβ inhibition, while % total proliferating cells was decreased with SB-431542 treatment (n = 3 mice). Transverse sections through the cut site of (**D, D', D'**) wild type injured tendon or (**E, E', E'**) TGFBR2[Scx] injured tendon stained for EdU and counterstained with DAPI. (**F**) Quantification of EdU and Scx[lin] overlays show no difference in Scx[lin] cell proliferation or total cell proliferation after injury with TGFBR2[Scx] deletion (n = 3 mice). A', B', D', E' are enlarged images from yellow boxed regions shown in A', B', C', D'. White arrows indicate EdU+, Scx[lin] cells. n.s. indicates p>0.1. Scale bars: 100 μm.

The online version of this article includes the following figure supplement(s) for figure 4:

**Figure supplement 1.** Proliferation in control, uninjured tendons is not affected by SB-431542 treatment or TGFBR2[Scx] deletion.

expansion through cell proliferation. In contrast, proliferation of non-Scx[lin] cells at d3 may depend on TGFβ signaling.

## Increased TGFβ ligand production in injured tendon depends on TGFβ signaling

Although Scx[lin] cells are not present in the gap space at d3, the region is not devoid of cells. At this time, we observed early accumulation of αSMA+ cells that are not derived from the Scx[lin] (*Howell et al., 2017*). Surprisingly, immunostaining for αSMA revealed that recruitment of αSMA+ cells at d3 was not affected by TGFβ inhibition or *Tgfbr2* deletion (*Figure 6A,B*). Transverse sections through the midsubstance gap space also confirmed that Scx[lin] cells were not yet detectable at d3 in any condition (not shown). The presence of αSMA+ cells within the gap space prior to Scx[lin] cell recruitment suggested that these cells may be a source of TGFβ ligands that signal to tenocytes for migration. Immunostaining for all three TGFβ isoforms showed comparable levels of signal between gap space of injured tendons and midsubstance of control tendons (*Figure 7A,B*). We therefore considered the possibility that ligand production may be regulated by TGFβ signaling and that impaired recruitment is due to reduction of TGFβ ligands with SB-431542 inhibition. However, immunostaining for TGFβ ligands showed equivalent staining intensity within the injury gap space of carrier- and SB-431542-treated limbs (*Figure 7A,B*). Differences were only observed in injured tendons at the cut

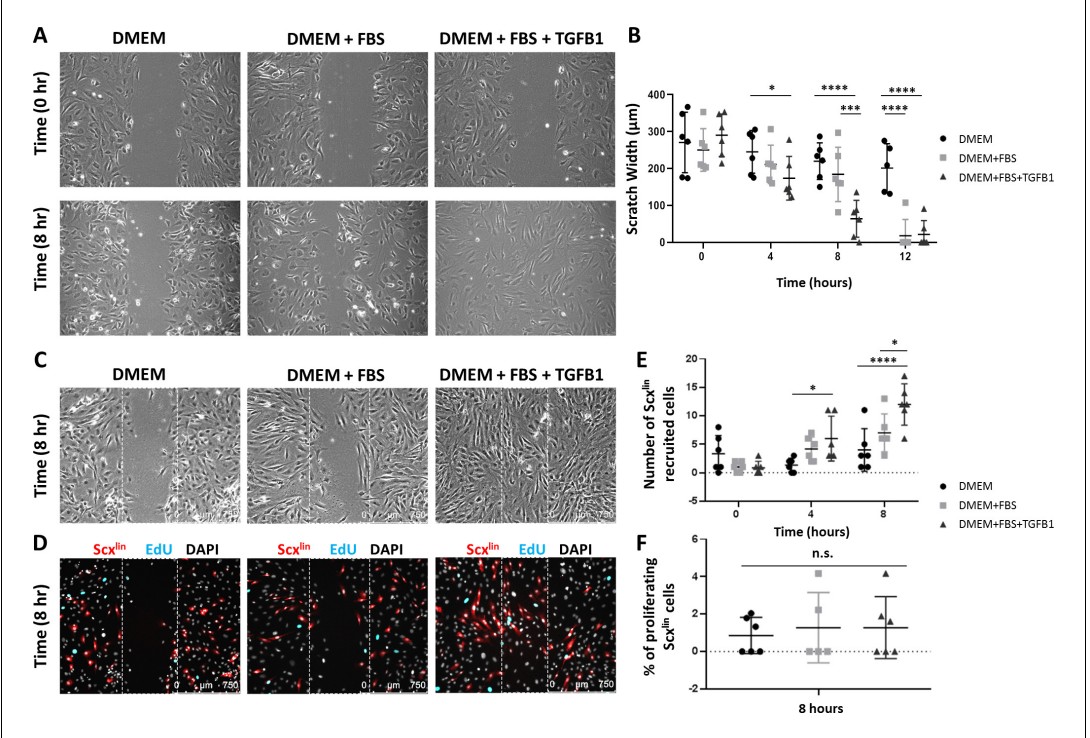

**Figure 5.** TGFβ enhances neonatal tenocyte migration in vitro. (A) Phase contrast images and (B) quantification of in vitro wound assay show accelerated closure with TGFB1 supplementation relative to DMEM and DMEM+FBS conditions (n = 6). (C) Phase contrast and (D) fluorescence images show enhanced (E) Scx[lin] cell migration with TGFB1 treatment. (F) No difference was observed in cell proliferation at 8 hr (n = 5–6). *p<0.05, ***p<0.001, ****p<0.0001. n.s. indicates p>0.1.

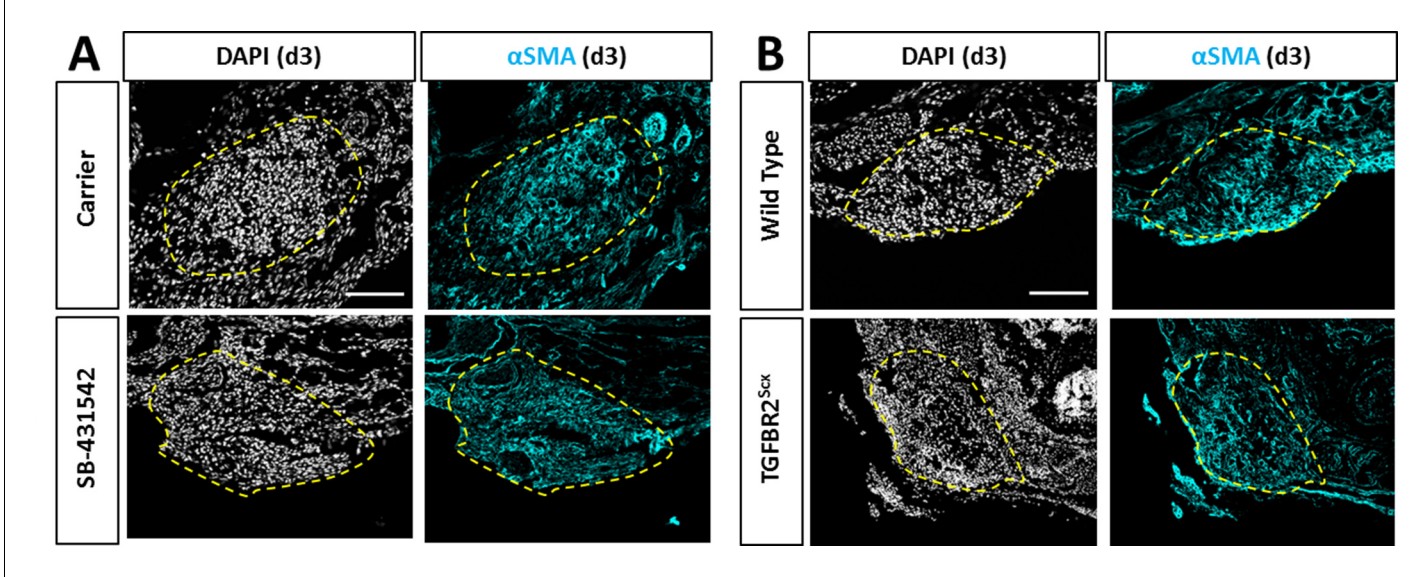

**Figure 6.** Recruitment of αSMA+ cells is not affected by TGFβ inhibition or TGFBR2[Scx] deletion. Transverse sections through the gap space at d3 showed abundant αSMA+ cells with (A) SB-431542 treatment or (B) TGFBR2[Scx] deletion at levels comparable to carrier-treated or wild type. Yellow dashed outlines highlight gap area formerly occupied by the Achilles tendon.
The online version of this article includes the following figure supplement(s) for figure 6:

**Figure supplement 1.** Increased detection of αSMA+ cells with SB-431542 treatment at d14 post-injury.

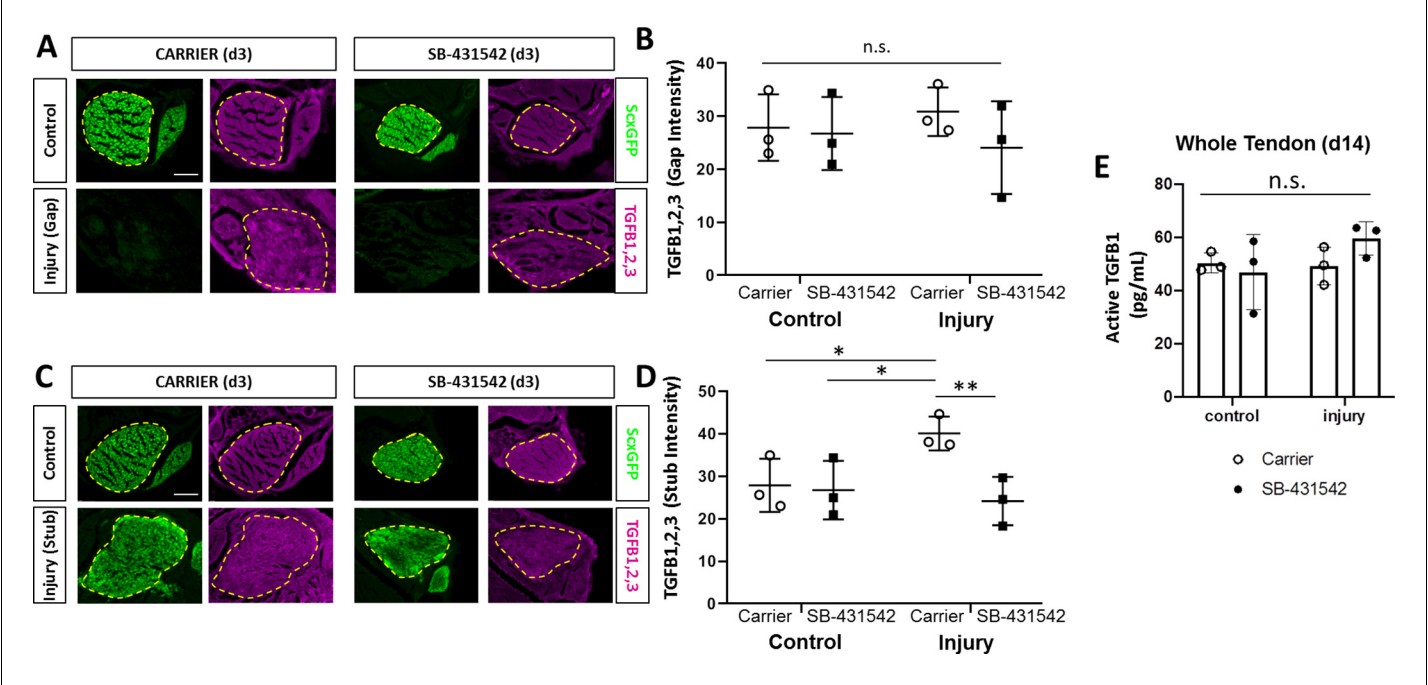

**Figure 7.** TGFβ ligand synthesis after injury is regulated by TGFβ signaling. (**A**) Transverse sections through the midsubstance tendon and neotendon regions at d3 immunostained with antibody against all three TGFβ isoforms. (**B**) Quantification of intensity levels show no difference in staining between groups. (**C**) Transverse sections through the tendon cut site regions at d3 immunostained with antibody against all three TGFβ isoforms. (**D**) Quantification of intensity levels show increased TGFβ ligands after injury in carrier-treated tendons that is no longer observed with SB-431542 treatment. Yellow dashed outlines indicate region of interest quantified. (**E**) TGFB1 protein quantification by ELISA showed no differences with injury or SB-431542 treatment at d14. *p<0.05 **p<0.01 n.s. indicates p>0.1. Scale bar: 100 μm.

site. We found increased ligand staining at the cut site of carrier-treated injured tendons relative to uninjured control, but this increase did not occur with SB-431542 inhibition (*Figure 7C,D*). Since the antibody does not distinguish between latent and active TGFβ ligands, we also quantified active TGFB1 protein by ELISA, but no differences were detected in whole tendon lysate by d14, regardless of injury or treatment (*Figure 7E*). These data suggest that TGFβ signaling is required for upregulation of TGFβ ligands with injury at d3, and that this is likely autonomously regulated in neonatal tenocytes.

## Non-Scx[lin] tenogenic cells also contribute to neotendon formation

Although αSMA+ cells are present at d3, immunostaining generally showed reduced αSMA+ cells by d14. However, more αSMA+ staining was observed with SB-431542 treatment relative to carrier (*Figure 6—figure supplement 1*). This may suggest that αSMA+ cells persist longer with TGFβ inhibition or maintain a progenitor or myofibroblastic phenotype. Since previous studies using *Acta2-CreERT2* (*Acta2* is the gene for αSMA) showed that adult αSMA-lineage (αSMA[lin]) cells of the paratenon surrounding tendons are resident progenitor cells for tendon (*Dyment et al., 2013*), we hypothesized that αSMA+ cells recruited into the gap space (which are not Scx[lin] at d3) may differentiate toward the tendon lineage and thereby turn off αSMA. Analysis of *Scx-GFP* expression in carrier-treated injured limbs indeed showed a population of non-Scx[lin], *Scx-GFP+* cells within the neotendon (*Figure 8A,B*). Comparison to contralateral non-injured controls indicated that incomplete recombination of Scx[lin] cells does not explain this phenomenon since recombination efficiency is ~96.4% in control tendons. Quantification of the non-Scx[lin] *Scx-GFP+* (*Scx-GFP* only) population showed fewer *Scx-GFP*-only cells in the neotendon after injury in SB-431542-treated mice (*Figure 8C–E*). There was a proportional decrease in DAPI+ cells, indicating that the reduction in *Scx-GFP* only cells was probably not due to failure of cells within the gap space to differentiate. Analysis of TGFBR2[Scx] mutants at d28 also showed the presence of *Scx-GFP*-only cells when Scx[lin] cells were minimally recruited (*Figure 8—figure supplement 1*).

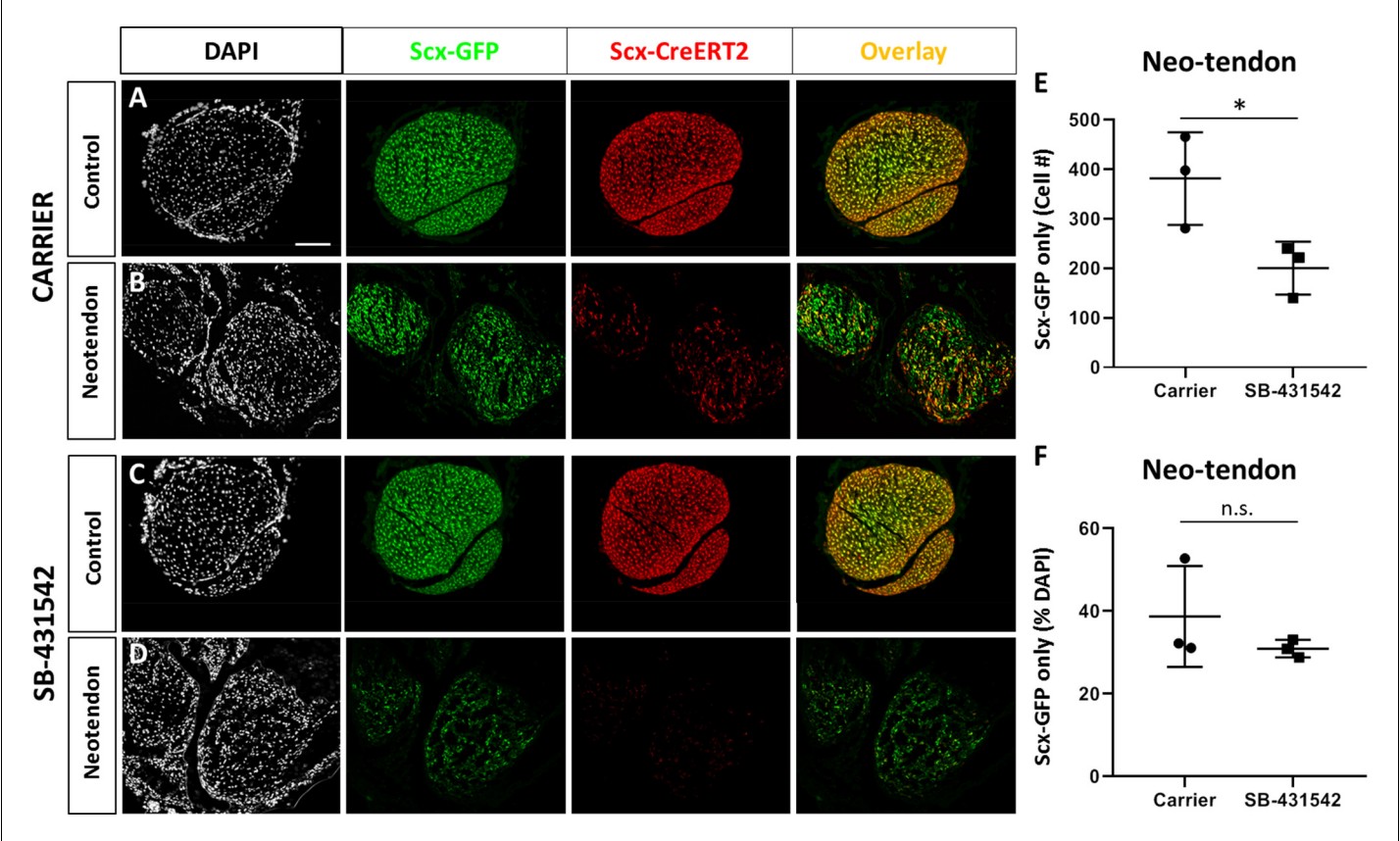

**Figure 8.** A population of *Scx-GFP*+, non-Scx[lin] cells are recruited after neonatal tendon injury. Transverse sections through the neotendon of control and injured limbs in (A, B) carrier-treated and (C, D) SB-431542-treated *Scx-CreERT2*; ROSA-Ai14; *Scx-GFP* mice. (E, F) Quantification of non-Scx[lin], *Scx-GFP*+ cells show reduction in cell number with SB-431542 treatment but not when normalized to total DAPI+ cells (n = 3 mice). *p<0.05. n.s. indicates p>0.1. Scalebar: 100 μm.

The online version of this article includes the following figure supplement(s) for figure 8:

**Figure supplement 1.** *Scx-GFP*+ cells are still detected in the absence of Scx[lin] recruitment in TGFBR2[Scx] mutants.

**Figure supplement 2.** Lineage tracing with *Acta2-CreERT2* show unexpected labeling in neonatal tenocytes.

**Figure supplement 3.** αSMA[lin] cells are recruited by d14 post-injury but do not account for all *Scx-GFP*+ cells.

To determine whether these non-Scx[lin], *Scx-GFP*+ cells were derived from αSMA+ cells, we labeled cells by tamoxifen administration at P2, P3 in transgenic *Acta2-CreERT2* mice. Analysis of transverse cryosections at P5 showed an unexpected amount of recombination in *Scx-GFP* tenocytes (*Figure 8—figure supplement 2*). Immunostaining for αSMA in control, uninjured tendons confirmed that neonatal tenocytes normally do not express αSMA. To determine recruitment and differentiation of αSMA[lin] cells, we labeled cells at P2, P3, injured tendons at P5 and harvested limbs at d14. Despite minimal αSMA immunostaining at d14 (*Figure 6—figure supplement 1*), we detected αSMA[lin] cells (TdTomato+) within the neotendon and the majority of αSMA[lin] cells appeared *Scx-GFP*-negative (*Figure 8—figure supplement 3*). In addition to TdTomato+/*Scx-GFP*- cells, we also identified TdTomato+/*Scx-GFP*+ cells and *Scx-GFP*+ cells in both carrier and SB-431542 treated neotendons (*Figure 8—figure supplement 3*).

Immunostaining results for αSMA at d14 suggested persistence of a progenitor phenotype with systemic TGFβ inhibition (*Figure 6—figure supplement 1*). To test whether tendon cell differentiation is affected, we determined tendon marker gene expression by real time qPCR at d3 and d14 post-injury. At d3, the tendon markers *Scx*, *Mkx*, and *Tnmd* were decreased in carrier-treated injured tendons compared to their contralateral uninjured controls (*Figure 9A*). Interestingly, inhibition of TGFβ signaling with SB-431542 also decreased tendon marker gene expression in uninjured control

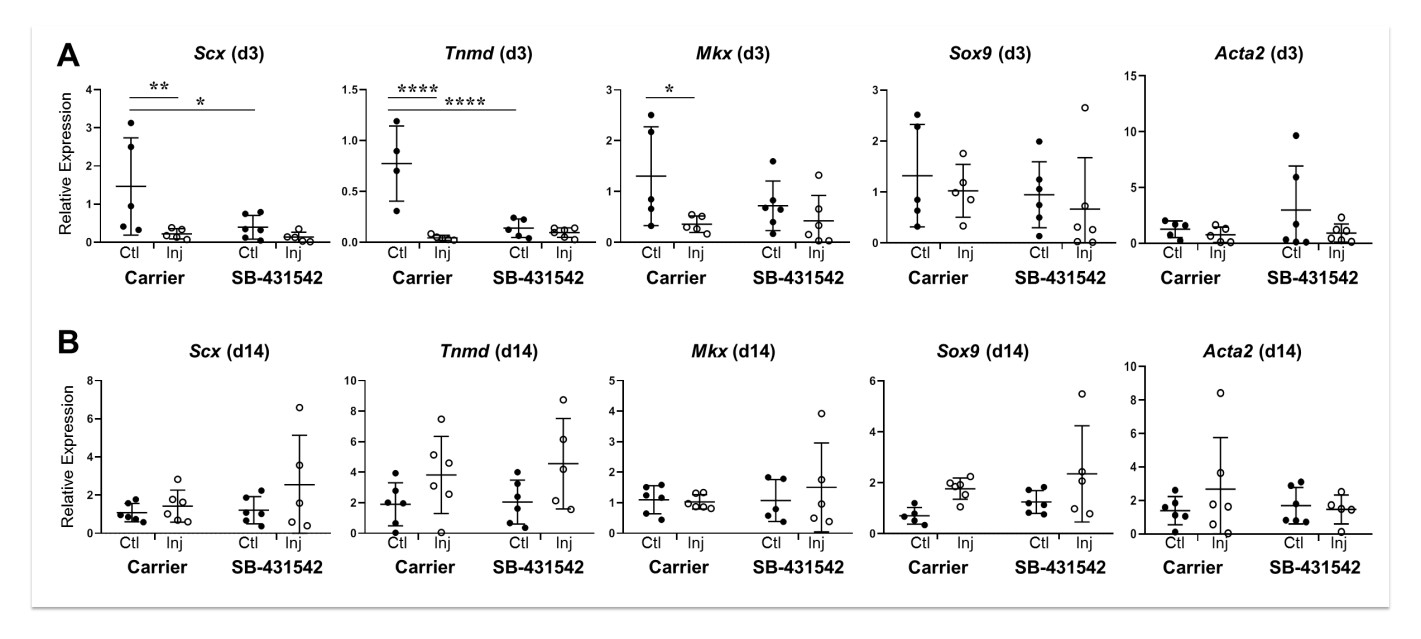

**Figure 9.** Tendon marker gene expression is not altered by TGFβ signaling inhibition at d14 post-injury. Real time qPCR analysis of tendons harvested at (A) d3 and (B) d14 from carrier-treated and SB-431542-treated animals (n = 4–6 mice). Tendon genes were decreased after injury in carrier-treated mice but not with SB-431542 treatment at d3. Differences were no longer detected by d14. *Sox9* and *Acta2* were not significantly different at either time point. *p<0.05, **p<0.01, ****p<0.0001.

tendons relative to carrier controls. By d14, tendon marker gene expression was unaltered regardless of treatment or injury (*Figure 9B*). Expression levels of *Sox9* and *Acta2* were not different across experimental conditions at either time point (*Figure 9A,B*). Collectively, these data indicate that despite defects in tenogenic cell recruitment, tendon marker gene expression and by extension tendon cell differentiation after injury was largely not affected by inhibition of TGFβ signaling.

## Discussion

In this study, we identified TGFβ-dependent and TGFβ-independent processes contributing to neonatal tendon regeneration. We found that early proliferation of Scx[lin] tenocytes and activation of aSMA+ cells do not depend on TGFβ signaling. However, proliferation of non-Scx[lin] cells, subsequent recruitment of tenogenic cells (from Scx[lin] and non-Scx[lin] sources), and functional restoration depend on TGFβ signaling (*Figure 10*). TGFβ signaling is a known regulator of many cellular processes, including cell proliferation, migration, and differentiation (*Shi and Massagué, 2003*). In tendon, TGFβ signaling is essential for embryonic tendon development as well as for the induction and maintenance of tendon cell fate (*Pryce et al., 2009*). However, after injury, TGFβ signaling is also known as a driver for fibrotic, scar-mediated healing and excessive TGFβ signaling results in tenocyte apoptosis (*Davies et al., 2016*; *Katzel et al., 2011*). In the context of tendon regeneration, it was therefore unclear whether TGFβ would be required for tendon differentiation or whether activation of TGFβ would drive fibrotic responses. Using our previously established model of neonatal tendon regeneration, we show here that TGFβ signaling is induced after injury and is required for the recruitment of neonatal tenocytes to the injury site. Since tenocyte proliferation after injury was not affected by inhibition of TGFβ signaling, we propose that tenocyte-mediated regeneration requires active migration of cells to bridge the gap space. This is further supported by in vitro data showing enhanced migration of neonatal tenocytes in the presence of TGFβ ligand and is consistent with several studies for other cell types (*Shi and Massagué, 2003*).

In addition to tenocytes, we also identified a second population of non-Scx[lin], *Scx-GFP+* cells that are recruited to the gap space. Inhibition of TGFβ signaling also resulted in reduced numbers of

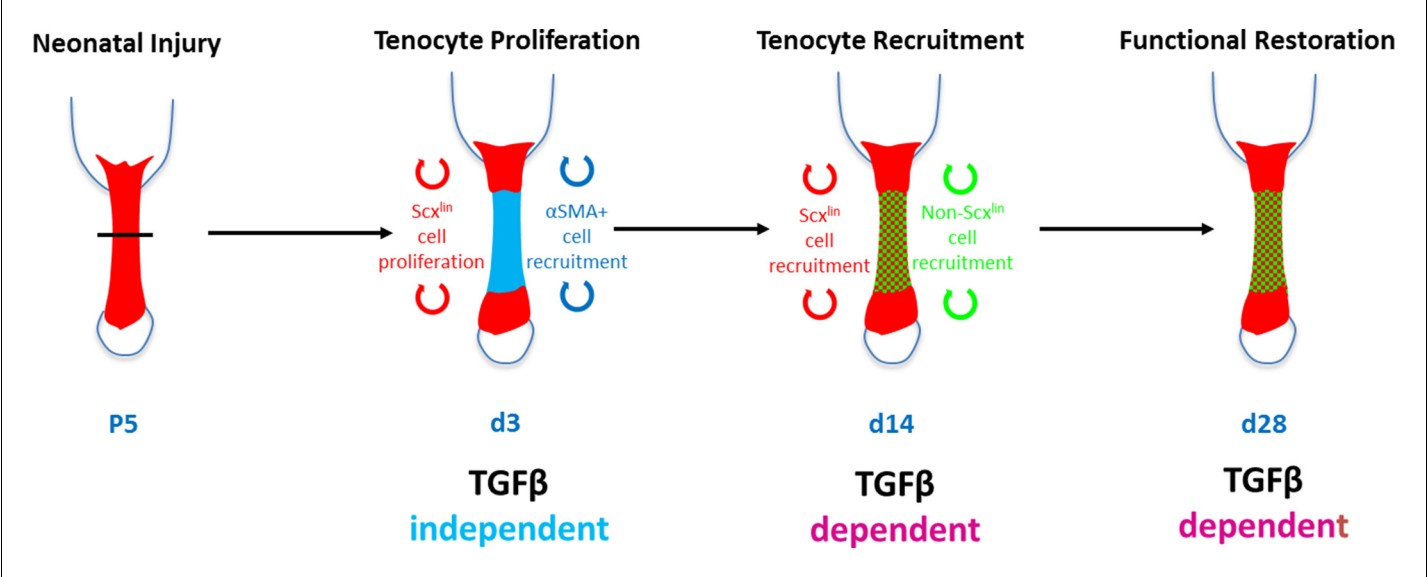

**Figure 10.** Requirement for TGFβ signaling in neonatal tendon regeneration. Schematic depiction of conceptual model of the TGFβ-dependent and TGFβ-independent cellular processes during neonatal tendon regeneration. While tenocyte proliferation and αSMA cell recruitment at d3 occur independently of TGFβ signaling, subsequent tenogenic cell recruitment and functional tendon tissue restoration requires TGFβ signaling.

these cells. One potential source of these cells may be the epitenon as it was previously proposed that tendon stem/progenitor cells reside in epitenon (*Dyment et al., 2014*; *Dyment et al., 2013*; *Gumucio et al., 2014*; *Mendias et al., 2012*; *Mienaltowski et al., 2013*). Although lineage tracing with *Acta2-CreERT2* showed restricted labeling in the epitenon/paratenon in adults (*Dyment et al., 2014*), we found considerable labeling in tenocytes at neonatal stages. After injury, αSMA[lin] cells were recruited into the neotendon, suggesting that this may be the source of non-Scx[lin], *Scx-GFP* cells. However, not all of the *Scx-GFP* cells in this line were derived from αSMA[lin] cells; therefore, the possibility of additional sources of tenogenic cells cannot be excluded. Alternatively, since *Acta2-CreERT2* labels only a sub-population of uninjured tenocyes, it may be that *Scx-GFP*-only cells in these mice are derived from the Scx[lin], non-αSMA[lin] population. Double labeling for αSMA[lin] and Scx[lin] cells would be required to test this hypothesis; however, this is not currently possible using existing genetic tools. Other sources of tenogenic cells include tendon-associated vasculature, as CD146+ pericytes have been identified near tendon (*Lee et al., 2015*). Since *Scx-GFP* may also be more broadly expressed compared to true *Scx* expression (*Best and Loiselle, 2019*; *Pryce et al., 2007*), the non-Scx[lin]/*Scx-GFP* and non-αSMA[lin]/*Scx-GFP* populations detected may also be a limitation of the transgenic reporter and not represent novel tenogenic cell populations. These issues can be addressed by using single cell RNASeq of sorted Scx[lin] and αSMA[lin] populations in future studies.

Unlike Scx[lin] cells, αSMA[lin] cells were present in the neotendon at d14 with SB-431542 treatment, consistent with immunostaining results showing considerable presence of αSMA+ cells at d3. However, quantitative measurements are required to determine whether proliferation or recruitment of these cells is affected by SB-431542 treatment. In addition to the αSMA[lin]/*Scx-GFP*+ cells occupying the neotendon at d14, we also identified αSMA[lin]/*Scx-GFP*- cells. These cells were largely negative for αSMA immunostaining at this timepoint, yet did not adopt a tenogenic phenotype. The role of this population in tendon regeneration is unclear, but they may function to support tenogenic cells, undergo tenogenesis themselves, or also adopt a fibrotic phenotype. Furthermore, it is unclear whether the αSMA[lin] cells in the gap space at d3 are the same ones observed in the neotendon at d14, or whether they represent a second wave of cell recruitment at later stages of the healing response. Additional experiments with quantitative measurements, temporal labeling, and localized ablation can be carried out in future studies to fully elucidate the dynamics of the αSMA[lin] populations, their interaction with the Scx[lin] population, and their roles in tendon regeneration and fibrosis. Despite impaired recruitment of tenogenic cells after TGFβ signaling inhibition, the expression of

tenogenic markers *Scx*, *Tnmd*, and *Mkx* were not different at d14. Identifying additional markers for tendon cell fate is the focus of ongoing studies.

We identified a potential source of TGFβ ligands in cells within the gap space, which may drive directional migration of the tenocytes from the stubs. After injury, there was an increase in TGFβ ligands in the tendon stub, which was suppressed by small molecule inhibition of TGFβ signaling. This suggests that initiation of TGFβ signaling results in positive feedback in tenocytes. We also observed a requirement for proliferation and recruitment of non-Scx^lin cells, but only with SB-431542 treatment and not in the TGFBR2^Scx mutant (which only targets TGFβ signaling in Scx^lin cells). This suggests that in the absence of Scx^lin cell recruitment, there is compensatory proliferation or recruitment of non-Scx^lin cells that also depends on TGFβ signaling. Other sources of TGFβs may be immune cells, which are known to secrete TGFβs. Of the three TGFβ isoforms, gene expression data suggested that the primary ligands driving neonatal regeneration may be TGFβs 1 and 3. Although *Tgfb1* showed bimodal upregulation pattern, *Tgfb3* was consistently upregulated after injury. Analysis of TGFB1 protein at d14 surprisingly showed a large change of total TGFB1 only, with no change in active TGFB1. TGFβs are typically secreted in a latent form bound to LTBPs in the extracellular matrix and release of TGFβs to its active form can be induced by proteases or mechanically (such as with transection injury) (*Maeda et al., 2011*). It is generally thought that ligand signaling can only occur with release of TGFβs from LTBP. However, recent studies using cryo-EM suggest that TGFβ activation can also occur without release from LTBP. This mechanism depends on interaction with the integrin αvβ8 (*Campbell et al., 2020*). Alternatively, it is also possible that signaling is mostly mediated by active TGFB3, which was not measured (validated ELISA kits to specifically detect mouse TGFB3 in tissue lysates are currently not available). During embryonic development, *Tgfb2* and *Tgfb3* are expressed in tendons and allelic deletion of these ligands results in increasing loss of tendons (*Kuo et al., 2008*; *Pryce et al., 2009*); in the context of injury, *Tgfb3* is expressed during regenerative fetal tendon healing in sheep while TGFB1 is associated with fibrotic adult tendon healing (*Beredjiklian et al., 2003*; *Kim et al., 2011*). Although this supports the notion that TGFβs 2 and 3 are pro-tenogenic relative to TGFB1, it is unclear whether the individual ligands actually can activate distinct healing or tenogenic responses. Additional research must therefore be carried out to elucidate their activities.

Although adult tendon healing was not determined in this study, it is well established that TGFβ signaling is elevated after adult injury and results in fibrotic scar formation (*Leask and Abraham, 2004*). Inhibition of TGFβ signaling, either with neutralizing antibodies or via $Smad3^{-/-}$ deletion attenuates fibrosis but fails to regenerate tendon structure or function (*Katzel et al., 2011*; *Kim et al., 2011*). We previously showed that adult tenocytes are largely quiescent after full transection injury with minimal cell proliferation or recruitment. The distinctive response of neonatal vs adult tenocytes to TGFβ may reflect differences in intrinsic potential (for example adult tenocytes are post-mitotic) or the activation of other signaling pathways that may interact with or modify TGFβ signaling. In addition to Smad signaling, TGFβs can also activate a number of non-Smad pathways (*Zhang, 2017*); there may be differences in downstream signaling between neonatal and adult tenocytes. Using an in vitro engineered tendon model, we previously showed that the tenogenic outcomes of TGFβ signaling did not depend on *Smad4* (*Chien et al., 2018*). Whether this finding is applicable in the context of in vivo injury remains to be determined.

## Materials and methods

### Key resources table

| Reagent type (species) or resource | Designation | Source or reference | Identifiers | Additional information |
|---|---|---|---|---|
| Genetic reagent (*M. musculus*) | Tg(Scx-GFP)1Stzr (ScxGFP) | Ronen Schweitzer (*Pryce et al., 2007*) | RRID:MGI:3717419 | |
| Genetic reagent (*M. musculus*) | Scx-CreERT2 | Ronen Schweitzer | N/A | |

*Continued on next page*

*Continued*

| Reagent type (species) or resource | Designation | Source or reference | Identifiers | Additional information |
|---|---|---|---|---|
| Genetic reagent (*M. musculus*) | Tg(Acta2-cre/ERT2)1Ikal(Acta2-CreERT2) | Ivo Kalajzic (*Grcevic et al., 2012*) | RRID:MGI:5461154 | |
| Genetic reagent (*M. musculus*) | Gt(ROSA) 26Sor^tm14(CAG-tdTomato)Hze (ROSA-Ai14) | Jackson Labs (*Madisen et al., 2010*) | Stock: 007908 RRID:MGI:3809524 | |
| Genetic reagent (*M. musculus*) | Tgfbr2^tm1.1Hlm (Tgfbr2^f/f) | Harold Moses (*Chytil et al., 2002*) | RRID:MGI:2384512 | |
| Chemical compound, drug | SB-431542 | Stemgent | Cat. # 04-0010-10 | |
| Antibody | Anti-alpha-SMA (mouse monoclonal) | Sigma-Aldrich | Cat. # A5228 RRID:AB_262054 | (1:100) |
| Antibody | Anti-TGFB1,2,3 MAb (Clone 1D11, mouse monoclonal) | R and D Systems | Cat. # MAB1835 RRID:AB_357931 | (1:50) |
| Antibody | Human Anti-TGFBR2 (goat polyclonal) | R and D Systems | Cat. # AF-241 | 5 ug/mL |
| Antibody | Donkey Anti-Rabbit Cy5 (rabbit polyclonal) | Jackson ImmunoResearch | Cat. # 711-175-152 RRID:AB_2340607 | (1:400) |
| Antibody | Strepdavidin Cy5 | Jackson ImmunoResearch | Cat. # 016-170-084 RRID:AB_2337245 | (1:400) |
| Antibody | Anti-phospho-Smad2/3 (rabbit polyclonal) | Cell Signaling | Cat. # 8828 RRID:AB_2631089 | (1:1000) |
| Antibody | Anti-phospho-p38 (rabbit monoclonal) | Cell Signaling | Cat. # 4511 RRID:AB_2139682 | (1:1000) |
| Antibody | Anti-GAPDH (mouse monoclonal) | Millipore Sigma | Cat. # MAB374 | (1:1000) |
| Antibody | Goat anti-rabbit IRDye (rabbit polyclonal) | Licor | Cat. # 926–68071 RRID:AB_2721181 | (1:10,000) |
| Antibody | Goat anti-mouse IRDye (mouse polyclonal) | Licor | Cat. # 926–32210 RRID:AB_621842 | (1:10,000) |
| Software, algorithm | ZEN Digital Imaging for Light Microscopy | Zeiss https://www.zeiss.com/microscopy/int/products/microscope-software/zen-lite.html | RRID:SCR_013672 | |
| Software, algorithm | ImageJ | ImageJ (http://imagej.nih.gov/ij/) | RRID:SCR_003070 | |
| Software, algorithm | Graphpad Prism | GraphPad Prism (https://graphpad.com) | RRID:SCR_015807 | |

## Experimental procedures

The following mouse lines were used: *Scx-GFP* tendon reporter (*Pryce et al., 2007*), *Scx-CreERT2* (generated by R. Schweitzer), *Acta2-CreERT2* (*Grcevic et al., 2012*), ROSA-Ai14 Cre reporter (*Madisen et al., 2010*), and *Tgfbr2^f/f* (*Chytil et al., 2002*). Lineage tracing and Cre deletion was

performed by delivering tamoxifen in corn oil to neonatal mice at P2 and P3 (1.25 mg/pup) (*Howell et al., 2017*). EdU was given at 0.05 mg 2 hr prior to harvest to label proliferating cells. Global TGFβ inhibition was carried out using the well-established small molecule inhibitor SB-431542 (10 mg/kg in 5% DMSO, intraperitoneal injection) which targets the TGFβ family type I receptors ALK 4/5/7 (*Hamilton et al., 2014*; *Inman et al., 2002*; *Laping et al., 2002*; *Lemos et al., 2015*; *Waghabi et al., 2009*). Daily injections of SB-431542 or carrier were administered from day 0–14 after injury. Full Achilles tendon transection without repair was carried out in neonates at P5, with male and female mice distributed evenly between groups. All procedures were approved by the Institutional Animal Care and Use Committee at Mount Sinai.

## Migration assay

Neonatal tenocytes were isolated from P7 pups by digestion in 1% collagenase type 1 (Cat. # LS004188, Worthington, Lakewood, NJ) and collagenase type 4 (Cat. # LS004188, Worthington, Lakewood, NJ) for 4 hr. Cells were expanded and maintained in high glucose DMEM (Cat. # 11965092, Life Technologies, Carlsbad, CA) with 10% fetal bovine serum (FBS, Life Technologies, Carlsbad, CA) and 1% PenStrep (Life Technologies, Carlsbad, CA). For the migration assay, cells were serum-starved for 24 hr and then maintained in DMEM only, DMEM+10% FBS, or DMEM+10% FBS+10 ng/mL TGFB1 (Cat. # 240-B, R and D Systems, Minneapolis, MN). A P200 tip was used scratch down the midline of every well. Phase contrast and fluorescence images were then taken every 4 hr for a total of 12 hr. Cell proliferation was measured by incubating cells with 0.05 mg EdU/ well for 30 min prior to harvest. EdU labeling was detected with the Click it EdU kit in accordance with manufacturer's instructions (Cat. # C10340, Life Technologies, Carlsbad, CA).

## Whole mount fluorescence imaging

Hind limbs were fixed in 4% paraformaldehyde (PFA, Cat. # 50-980-495, Fisher Scientific, Waltham, MA) overnight at 4°C and skin removed to expose the Achilles tendon. Whole mount images of the posterior limbs were captured using a Leica M165FC stereomicroscope with fluorescence capabilities. Exposure settings were maintained across limbs.

## Immunofluorescence and microscopy

After sacrifice, limbs were fixed in 4% PFA for 24 hr at 4°C, decalcified in 50 mM EDTA for 1–2 weeks at 4°C, then incubated in 5% sucrose (1 hr) and 30% sucrose (overnight) at 4°C. Limbs were then embedded in optimal cutting temperature medium (Cat. # 23–730, Fisher Scientific, Waltham, MA) and 12 um transverse cryosections obtained. Immunostaining was carried out with primary antibodies against αSMA (Cat. # A5228, Sigma, St. Louis, MI), TGFB1,2,3 ligands (Cat. # MAB1835, R and D Systems, Minneapolis, MN), TGFBR2 (Cat. # AF-241, R and D Systems, Minneapolis, MN) and secondary detection with antibodies conjugated to Cy5 (Cat. # 711-175-152; 016-170-084, Jackson ImmunoResearch, West Grove, PA). EdU labeling was detected with the Click it EdU kit in accordance with manufacturer's instructions (Cat. # C10340, Life Technologies, Carlsbad, CA). Fluorescence images were acquired using the Zeiss Axio Imager with optical sectioning by Apotome or using the Leica DMB6 microscope. Cell quantification was performed in Zeiss Zen or Image J software on transverse cryosection images. Immunofluorescence measurements were acquired in ImageJ by measuring average intensity of grayscale images. All images for quantifications were taken at the same exposure and image manipulations applied equally across samples.

## RNA isolation, reverse transcription, and qRT-PCR

Trizol/chloroform extraction was used to isolate RNA from dissected tendons. cDNA was then synthesized via reverse transcription using the SuperScript VILO master mix (Cat. # 11755050, Invitrogen, Carlsbad, CA). Gene expression was assessed by qRT-PCR using SYBR PCR Master Mix (Cat. # 4309155, Thermo Fisher, Waltham, MA) and calculated using the standard curve method or the $2^{-\Delta\Delta Ct}$ method relative to carrier-treated control tendons. The housekeeping gene, *Gapdh,* was used to normalize gene expression. Primer sequences for TGFβ-related molecules are listed in *Supplementary file 1*. All other primers were previously described (*Howell et al., 2017*).

## Protein extraction

Tendons were frozen in liquid nitrogen and pulverized in a Geno/Grinder at 1,500 rpm for 90 s. Pulverized tendons were resuspended in 150 µL T-PER tissue protein extraction reagent (Thermo Scientific, 78510), supplemented with cOmplete protease inhibitor (Roche, 04693159001), and PhosSTOP phosphatase inhibitor (Roche, 4906845001) according to the manufacturer's instructions. To allow for lysis, tendons were rotated end-over-end for 1 hr at 4°C and then ultrasonicated at 10 A for 30 s. Finally, samples were cleared at 10,000 x g for 5 min and the supernatant was collected for western blot analysis and ELISA. Protein concentration was determined using a Bradford assay (Thermo Scientific, PI23238) and a NanoDrop spectrophotometer (Thermo Scientific).

## Western blotting

Protein extracts were incubated with 5x reducing sodium dodecyl sulfate and polyacrylamide gel electrophoresis (SDS-PAGE) buffer for 5 min at 95°C. Equal volumes of samples were loaded onto 7.5% polyacrylamide gels and separated using a Mini-PROTEAN Tetra Vertical Electrophoresis system (Bio-Rad) at 80 V for 20 min followed by 120 V for 1 hr. After separation of the proteins, gels were transferred to poly-vinylidene difluoride (PVDF) membranes (Immobilon-FL, IPFL00010) using a Mini Trans-Blot Cell (Bio-Rad) at 70 V for 90 min. Following protein transfer, membranes were blocked for 1 hr at RT with 5% bovine serum albumin (BSA) in Tris-buffered saline (TBS). Blots were then incubated with rabbit polyclonal antibody detecting phospho-Smad2/3 (Cell Signaling Technology, 8828), monoclonal antibody detecting phospho-p38 (Cell Signaling Technology, 4511) and a mouse monoclonal GAPDH antibody (Millipore Sigma, MAB374) as loading control, all diluted 1:1000 in 5% BSA in TBS with 0.1% TWEEN-20 (TBS-T) overnight at 4°C. Blots were then washed with TBS-T three times for 5 min and incubated with IRDye 680RD goat anti-rabbit and IRDye 800CW goat anti-mouse secondary antibodies (Licor, 926–68071 and 926–32210) diluted 1:10,000 in 5% BSA in TBS-T for 2 hr at RT. Blots were washed with TBS-T three times for 5 min and TBS for 5 min one time and imaged using the Licor Odyssey imaging system. Fluorescent intensity of individual bands was quantified with ImageJ software and normalized to GAPDH.

## ELISA for TGFB1 quantification

Undiluted protein extracts were assayed with the TGFB1 Quantikine ELISA Kit (R and D Systems, DB100B) according to the manufacturer's instructions. To assess total TGFB1, latent TGFB1 was activated by incubating samples with 1N HCl for 10 min and neutralizing with 1.2N NaOH in 0.5M HEPES. Samples were quantified using a SpectraMax microplate reader and SoftMax Pro software against a standard curve (0–1000 pg/mL TGFβ1).

## Gait analysis

Mice were gaited at 10 cm/s for 3–4 s using the DigiGait Imaging System (Mouse Specifics Inc, Quincy, MA). A high-speed digital camera was used to capture forelimb paw positions and parameters previously established for mouse Achilles tendon injury were then extracted (Howell+, Sci Rep, 2017). All parameters were normalized to Stride length to account for differences in animal size and age.

## Biomechanical testing

Tensile testing was performed in PBS at room temperature using custom 3D printed grips to secure the calcaneus bone and Achilles tendon (*Abraham et al., 2019*). Tendons were preloaded to 0.05N for ~1 min followed by ramp to failure at 1 %/s. Structural properties were recorded; since cross-sectional area could not be accurately measured due to the small size of the tissues, material properties were not analyzed.

## Statistical analysis

Quantitative results are presented as mean ± standard deviation. Two way ANOVA was used for comparisons with two independent variables (injury and TGFβ inhibition); where significance was detected, posthoc testing was then carried out (Graphpad Prism). All other quantitative analyses were analyzed using Students t-tests. Significant outliers were detected using Grubb's test (Graphpad Prism). Sample sizes for gait and mechanical properties quantification were calculated from

power analyses with power 0.8% and 5% type I error. Samples sizes for other quantitative data were used based on previous data from the lab and published literature. Significance was determined at $p < 0.05$.

## Acknowledgements

This work was supported by NIH/NIAMS R01AR069537 to AHH, F31AR073626 to DK, and R01AR070748 NIH/NIAMS to DH.

## Additional information

### Funding

| Funder | Grant reference number | Author |
|---|---|---|
| National Institutes of Health | R01AR069537 | Alice H Huang |
| National Institutes of Health | F31AR073626 | Deepak A Kaji |
| National Institutes of Health | R01AR070748 | Dirk Hubmacher |

The funders had no role in study design, data collection and interpretation, or the decision to submit the work for publication.

### Author contributions

Deepak A Kaji, Conceptualization, Data curation, Formal analysis, Funding acquisition, Validation, Investigation, Visualization, Methodology; Kristen L Howell, Data curation, Formal analysis, Investigation, Methodology; Zerina Balic, Data curation, Formal analysis, Methodology; Dirk Hubmacher, Conceptualization, Resources, Data curation, Formal analysis, Supervision, Funding acquisition, Investigation, Methodology, Writing - review and editing; Alice H Huang, Conceptualization, Resources, Formal analysis, Supervision, Funding acquisition, Investigation, Methodology, Project administration

### Author ORCIDs

Deepak A Kaji https://orcid.org/0000-0002-0470-3219
Dirk Hubmacher https://orcid.org/0000-0003-1569-9451
Alice H Huang https://orcid.org/0000-0002-5037-6829

### Ethics

Animal experimentation: This study was performed in strict accordance with the recommendations in the Guide for the Care and Use of Laboratory Animals of the National Institutes of Health and all procedures approved by the institutional animal care and use committee (IACUC) at Mount Sinai (IACUC-2014-0031).

### Decision letter and Author response

Decision letter https://doi.org/10.7554/eLife.51779.sa1
Author response https://doi.org/10.7554/eLife.51779.sa2

## Additional files

### Supplementary files

- Supplementary file 1. Primer sequences for real time qPCR.

- Transparent reporting form

### Data availability

All data analyzed in this study are included in the manuscript.

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
