## [Decision Letter]

**Acceptance summary:**

We appreciate your thoughtful response. The paper represents a novel addition to existing literature on tendon biology. It delineates the molecular basis for neonatal tendon regeneration, And particularly highlights the role of TGFβ signaling. Through a combination of targeted gene deletion, small molecule inhibition, and lineage tracing, the publication identifies TGFβ-dependent and -independent cellular mechanisms involved in tendon regeneration.

**Decision letter after peer review:**

Thank you for submitting your article "TGFβ signaling is required for tenocyte recruitment and functional neonatal tendon regeneration" for consideration by *eLife*. Your article has been reviewed by three peer reviewers, and the evaluation has been overseen by Mone Zaidi as the Reviewing Editor and Clifford Rosen as the Senior Editor. The following individuals involved in review of your submission have agreed to reveal their identity: Mei Wan (Reviewer #1); Alayna E Loiselle (Reviewer #2); Nathanial Dyment (Reviewer #3).

The reviewers have discussed the reviews with one another and the Reviewing Editor has drafted this decision to help you prepare a revised submission.

Summary:

The studies potentially reveal a new molecular pathway that regulates neonatal tendon regeneration. However, there are several significant concerns with the approach and methodology. The reviewers consider it necessary to validate the labeling specificity of the ScxCreERT2/RosaT mice, document an inhibitory efficiency of the small compound/TβR2 KO on TGFβ signaling; and evaluate cell proliferation.

Essential revisions:

1) There is a paucity of evidence for pharmacologic and genetic inhibition of the Tgfb pathway, particularly at the dose of the chemical inhibitor that was used. The specificity of labeling in reporter ScxCreERT2/RosaT mice needs further validation, particularly in relation to possible leakiness in the absence of tamoxifen. Exclusion of false positivity is essential. Furthermore, it is unclear whether non-Scx^lin^, ScxGFP+ cells are also labeled by the aSMACreERT2 mouse during the healing process.

2) Another concern relates to the measurement of functionally active Tgfb, and expressing it as a ratio of total Tgfb. This is because TGFβ is expressed with the latency-associated protein (LAP), rendering it inactive by masking the extracellular matrix (ECM) in many different tissues, including tendons. TGFβ does not have function unless it is activated. Therefore, the mRNA level of TGFβ in tendon tissue (Figure 1) does not reflect the activation of TGFβ signaling. Active TGFβ/ total TGFβ ratio should be measured.

3) The Westerns showing phosphorylated Smad2/3 need to be replicated and quantitated to establish target engagement and inhibition.

4) To enhance clarity and firm up conclusions, new data relating to proliferation and migration are required. An in vitro cell proliferation assay is recommended. It is also essential that the authors quantitate proliferating cells and present these as total % of proliferative cells. A scratch assay in parallel to assess migration would be a worthwhile addition. Finally, the authors should examine for migration in TBR2 deficient tenocytes to further support tenocyte-intrinsic effects.

Other points to be addressed:

1) Tmx labelling should be until ~D7-10 post-injury to confirm that the ScxGFP and ScxAi4 cells overlap, given that the ScxGFP construct has a broader expression pattern than endogenous Scx.

2) There is a question on whether diminished recruitment of ScxGFP cells also occurs in TβR2^Scx^ mice? This may require more elaborate studies.

3) The apparent discrepancy between RosaT-labeled cells in Figures 3 and 4 needs to be addressed. "Little RosaT signal was detected in SB-431542-treated limbs" in Figure 3B and C and "few Scx^lin^ tenocytes were detected in TβR2^Scx^ mutant tendons" in Figure 3F and G. However, there are many red color Scx^lin^ tenocytes were shown in SB-431542-treated (Figure 4B' and B') and TβR2^Scx^ mutant tendons. What is the reason for this discrepancy?

4) In Figure 7, TGFβ ligand were decreased in SB-431542 treated injured mice. The authors concluded that TGFβ signaling is required for upregulation of TGFβ ligands with injury. This conclusion is confusing.

5) The authors have previously shown insufficient tenocyte recruitment in adult healing, but appropriate recruitment in neonates- is this a function of size/distance since multiple other models show bridging tissue or is this an AT specific finding?

6) Figure 3D: what do the authors attribute the massive variation in the # of TdTomato+ cell recruitment in inhibitor treated animals to? These data should also be analyzed as percentage of total cells in the bridging tissue. This will demonstrate whether inhibition affects all cell recruitment or specifically Scx^lin^.

7) Figure 4: In addition to normalization to carrier and WT for Edu labelling, the total % of proliferative cells would be informative about effects of Tgfb inhibition or deletion on general cellular dynamics.

8) Supplementary Figure 3/Figure 7: How was intensity of TGFb ligand staining quantified? Since I believe the authors main point is that there are spatial differences it would be appropriate to have these data on the same graph, and to have representative images from the gap space. In addition, what are the two different stains (purple/green) in Figure 7A? Ideally, quantitative analyses of ligand levels via ELISA would be most convincing, but I recognize the technical challenges with this.

9) TGFb is known to stimulate proliferation of cells in culture. It appears that the cells were not serum starved just prior to the scratch assay or a proliferation inhibitor (e.g., Mitomycin C) was not added in one of the groups. Therefore, it's difficult to determine whether the changes were due to migration or proliferation. It would also be a nicer parallel to the in vivo studies to do the scratch assay with the Tgfbr2 cKO cells.

10) It is recommended not to call aSMA+ cells in this paper only myofibroblasts. Some of them are likely myofibroblasts but not all. Even though aSMA is a marker of a myofibroblast, all mesenchymal cells that express aSMA are not myofibroblasts. aSMA is expressed during expansion of progenitor cell populations that contribute to tissue healing. These cells go on to differentiate into ScxGFP+ cells, osteocytes, fibrocartilage, and odontoblasts. Myofibroblasts, on the other hand, are a more mature cell type. The key defining features of myofibroblasts are the de novo development of in vivo stress fibers and contractile forces. Stress fibers containing aSMA produce higher contractile forces, which is why myofibroblasts express aSMA. The fact that aSMA expression was not inhibited at this early stage by SB-431542, despite TGFb being a potent inducer of myofibroblasts (as stated in the third paragraph of the Discussion), also indicates that these cells may not be myofibroblasts.

11) It is suggested that the authors add uninjured limbs to the main figure instead of in Figure 2—figure supplement 1. Panels A and B have the uninjured limbs so it would balance the figure nicely.

12) Figure 4C – it would help to indicate on the graph that proliferation of the Scx^lin^ cells is reported.

13) Figure 6—figure supplement 1 – Was aSMA staining higher at D14 in SB-431542-treated group? This could further support that healing is diminished/delayed in the treated group.

14) The titles of Figures 7 and 8 are the same.

---

## [Author Response]

Essential revisions:1) There is a paucity of evidence for pharmacologic and genetic inhibition of the Tgfb pathway, particularly at the dose of the chemical inhibitor that was used. The specificity of labeling in reporter ScxCreERT2/RosaT mice needs further validation, particularly in relation to possible leakiness in the absence of tamoxifen. Exclusion of false positivity is essential.

Pharmacologic inhibition

SB-431542 is a very well established TGFβ inhibitor used in many in vivo mouse studies at doses ranging from 0.17 mg/kg to 10mg/kg, with various injection regimens such as oral gavage, single IP injection, or daily IP injection (see references listed at end of response: (Alyoussef, 2018; Araujo-Jorge et al., 2012; Bayomi et al., 2013; Chowdhury et al., 2015; Koh et al., 2015; Lemos et al., 2015; Lu et al., 2018; Mercado-Gomez et al., 2014; Miao et al., 2014; Mohamed et al., 2017; Mordasky Markell et al., 2010; Pulli et al., 2015; Sato et al., 2015; Schindeler et al., 2010; Shi et al., 2017; Tanaka et al., 2010; Waghabi et al., 2007; Wei et al., 2010; Zhang et al., 2016; Zhou et al., 2019)). We chose our dosing regimen based on this prior literature and used 10 mg/kg daily IP injections as the most commonly used, highest dose, and therefore most likely to be effective. To determine the effect of the drug on TGFβ signaling, we carried out western blotting for pSmad2/3 with carrier and SB-431542 treatment. We were surprised not to see a difference between injured tendons (carrier vs. inhibitor), however there was a significance increase in uninjured control tendons with inhibition. This suggested that the systemic level of pSmad2/3 is generally higher in those mice and we therefore normalized to the contralateral control. Once normalized, there was a ~67% reduction in pSmad2/3 with SB-431542 treatment (new Figure 2—figure supplement 1).

This difference in pSmad2/3 in control tendons could indicate a potential feedback mechanism with SB-431542 inhibition or potential compensation by other TGFβ superfamily molecules. We also tested whether this result was specific to pSmad2/3 by assaying phospho-p38 protein (a non-Smad mediator of TGFβ signaling) and found no difference with injury or with inhibitor treatment, either normalized or un-normalized. This is expected as SB-431542 should not affect ERK or p38 signaling, based on in vitro studies (see (Inman et al., 2002). There is also a question of whether we chose the right time (2 hours after last injection) to harvest the tendons. There is evidence that TGFβ signaling is cyclic (Zi et al., 2012); the compound may therefore only inhibit at certain durations per day. Although our pSmad2/3 results were somewhat unexpected, we do see a relative inhibition with SB-431542 and many of our results (the functional assays, recruitment, total cells) point to an effect of the inhibitor on healing. Materials and methods, Results subsection “TGFβ signaling is required for functional regeneration”, and Discussion have been updated with a new Figure 2—figure supplement 1.

Genetic deletion

The original Tgfbr2^Scx^ samples were extremely hard to come by. We lost many litters (representing perhaps ~200 mice) after tamoxifen treatment and the samples included here were the few that survived. Outcrossing the line did not help (we tried twice) and after years of attempted experiments, we eliminated the line in early 2019, prior to the initial submission of this manuscript. While we were therefore not able to generate new mice for experiments, we examined remaining cryosections and tried immunostaining for pSmad2/3 (did not work) and immunostaining for TGFBR2. We now include the TGFBR2 results showing reduced immunostaining of TGFBR2 in Tgfbr2^Scx^ mutants with tamoxifen treatment (new Figure 3—figure supplement 2, subsection “TGFβ signaling in neonatal tenocytes is required for cell recruitment after injury”).

The specificity of labeling in reporter ScxCreERT2/RosaT mice needs further validation, particularly in relation to possible leakiness in the absence of tamoxifen.

Analysis of ScxCreERT2/RosaT/ScxGFP mice at 1 year of age without any tamoxifen injections showed no presence of RosaT+ cells suggesting that there is no leakiness. We chose the oldest age maintained in our colony to span the widest range of time for leakiness to possibly occur. This is now included as Figure 3—figure supplement 1 (subsection “TGFβ signaling in neonatal tenocytes is required for cell recruitment after injury”.

Furthermore, it is unclear whether non-Scx^lin^, ScxGFP+ cells are also labeled by the aSMACreERT2 mouse during the healing process.

This is a very interesting point. To truly address this, we would likely need to lineage trace both Scx^lin^ and aSMA^lin^ cells within the same mouse, which is currently impossible since these are both CreERT2 lines. However, if all ScxGFP cells are from the aSMA^lin^, this would point to the epitenon/paratenon as a likely second source of cells. To test this, we generated aSMACreERT2/RosaT/ScxGFP pups with P2, P3 tamoxifen labeling and P5 injury, with and without injection of SB-431542 (n=2/group although 1 mouse ended up ScxGFP-negative in the untreated group and was not useful). All pups at d14 showed recruitment of RosaT+ cells into the gap space, regardless of SB-431542 treatment. This is consistent with our aSMA immunostaining showing that aSMA cells were not affected by SB-431542 treatment at d3. We found aSMA^lin^ cells comprised a majority of non-ScxGFP cells and a proportion (but not all) of ScxGFP+ cells (new Figure 8—figure supplement 3). Since aSMA labels some of the original tenocytes (Figure 8—figure supplement 2) we still cannot completely exclude the possibility that aSMA^lin^+/ScxGFP+ cells are also Scx^lin^. The Results and Discussion have been revised accordingly with new Figure 8—figure supplement 3. (See subsection “Non-Scx^lin^ tenogenic cells also contribute to neotendon formation” and Discussion).

2) Another concern relates to the measurement of functionally active Tgfb, and expressing it as a ratio of total Tgfb. This is because TGFβ is expressed with the latency-associated protein (LAP), rendering it inactive by masking the extracellular matrix (ECM) in many different tissues, including tendons. TGFβ does not have function unless it is activated. Therefore, the mRNA level of TGFβ in tendon tissue (Figure 1) does not reflect the activation of TGFβ signaling. Active TGFβ/ total TGFβ ratio should be measured.

We measured active and latent TGFβ1 using ELISA and show increased levels of total TGFβ1 with injury and no change in active TGFβ1. Although it was long thought that TGFβ cannot signal without release from the latent complex, an interesting paper in Cell recently showed that latent TGFβ may also be able to signal by binding through the integrin αvβ8 (Campbell et al., 2020). It is also possible that most of the signaling is mediated by TGFβ3, which we were not able to assay since the mouse TGFβ2 and TGFβ3 ELISA kits are either not commercially available or not validated for tissue lysates (as discussed with the manufacturer for the TGFβ3 kit). We do identify increases in pSmad2/3 signaling however, which indicates active signaling with injury. We now include this Cell reference and updated Materials and methods, Results subsection “TGFβ signaling is activated after neonatal injury”, and Discussion.

3) The Westerns showing phosphorylated Smad2/3 need to be replicated and quantitated to establish target engagement and inhibition.

Western blots for pSmad2/3 are now included using newly collected samples and quantified as requested (updated Figure 1, subsection “TGFβ signaling is activated after neonatal injury”).

4) To enhance clarity and firm up conclusions, new data relating to proliferation and migration are required. An in vitro cell proliferation assay is recommended. It is also essential that the authors quantitate proliferating cells and present these as total % of proliferative cells. A scratch assay in parallel to assess migration would be a worthwhile addition.

We agree with the reviewer that the proliferative effects of TGFβ may complicate our interpretation of the scratch assay. First, the cells were indeed serum-starved prior to the assay and the methods are now updated to reflect this (we apologize for omission). However, EdU staining did show some degree of proliferation occurred despite serum-starving. Therefore, we now include the requested experiment repeating the scratch assay and quantifying proliferation to determine whether proliferation and migration were in any way coupled. Our results show enhanced gap space closure with the addition of TGFβ1 without a significant difference in proliferation (updated Figure 5, subsection “TGFβ signaling is required for tenocyte migration but not proliferation”).

Finally, the authors should examine for migration in TBR2 deficient tenocytes to further support tenocyte-intrinsic effects.

As explained above, the Tgfbr2 floxed line is no longer available in our colony. To determine tenocyte intrinsic effects in the culture model, we therefore labeled cells using ScxCreERT2 and analyzed the behavior of the labeled cells in the dish. Quantification showed enhanced migration capacity of Scx^lin^ cells in the presence of TGFβ1 (updated Figure 5, subsection “TGFβ signaling is required for tenocyte migration but not proliferation”).

Other points to be addressed:1) Tmx labelling should be until ~D7-10 post-injury to confirm that the ScxGFP and ScxAi4 cells overlap, given that the ScxGFP construct has a broader expression pattern than endogenous Scx.

We were not able to carry out this experiment due to lab closure, but the broader expression pattern of ScxGFP is an excellent and important point; we have now included this as a Discussion point. This may also reflect incomplete recombination (although we do see ~96% of ScxGFP+ tenocytes in the contralateral control tendon is recombined).

2) There is a question on whether diminished recruitment of ScxGFP cells also occurs in TβR2^Scx^ mice? This may require more elaborate studies.

Unfortunately, many of the Tgfbr2^Scx^ samples collected did not have ScxGFP (this includes all of the d14 samples). We had 2 mutant samples collected at d28 (which were not included in the original manuscript) that showed the presence of ScxGFP cells despite minimal Scx^lin^ cell recruitment. Given our new results with aSMACreERT2, it is likely this population (at least in part) comes from aSMA^lin^ cells. This is now included as Figure 8—figure supplement 1 and the Results subsection “Non-Scx^lin^ tenogenic cells also contribute to neotendon formation” revised.

3) The apparent discrepancy between RosaT-labeled cells in Figures 3 and 4 needs to be addressed. "Little RosaT signal was detected in SB-431542-treated limbs" in Figure 3B and C and "few Scx^lin^ tenocytes were detected in TβR2^Scx^ mutant tendons" in Figure 3F and G. However, there are many red color Scx^lin^ tenocytes were shown in SB-431542-treated (Figure 4B' and B') and TβR2^Scx^ mutant tendons. What is the reason for this discrepancy?

Figure 4 showed a section at the injured edge of the tendon stub at d3 where we see the original tenocytes (highly labeled) proliferating at d3. Figure 3 is from the midsubstance (former gap space) at d14 where we quantify recruitment (all labeled cells here would have had to migrate into the space). We have clarified this in the revised text (adding ‘midsubstance region’ and ‘injured tendon edge’ to describe locations of transverse sections (subsection “TGFβ signaling in neonatal tenocytes is required for cell recruitment after injury”.

4) In Figure 7, TGFβ ligand were decreased in SB-431542 treated injured mice. The authors concluded that TGFβ signaling is required for upregulation of TGFβ ligands with injury. This conclusion is confusing.

It is decreased relative to the carrier injured group but not to the non-injured controls. We interpret this to mean that normally after injury, there is an increase in TGFβ ligands at d3 at the cut site. With TGFβ inhibition, the increase is not observed (the SB-431542 injured group is not different from the carrier or SB-431542 non-injured groups). While it would also be helpful to have TGFβ1 ELISA data from the inhibitor treated cohort at d3 to support these data, we were not able to generate additional mice at this time. In addition, since the increase is specific to a localized region of the tendon, it is not clear the difference would be detectable since whole tendon lysates are combined and used for ELISA.

5) The authors have previously shown insufficient tenocyte recruitment in adult healing, but appropriate recruitment in neonates- is this a function of size/distance since multiple other models show bridging tissue or is this an AT specific finding?

We measured the gap space normalized to the original geometry of the contralateral tendon and found there is no difference in relative gap space between neonates and adults (indeed, relative gap space normalized to length may be slightly larger in neonates). Since the current study does not include adult injury, we only show these data as Author response image 1. We posit that the transection injury without repair creates a critical size defect in adults and adult tenocytes are not able to bridge this gap while neonatal tenocytes are able. Elegant studies from Dr. Loiselle’s lab observed successful bridging of adult tenocytes of flexor tendons when tendon ends are sutured, suggesting there may be a gap distance threshold for adult tenocyte recruitment. Note that although the adult gap space is not occupied by Scx^lin^ tenocytes, there IS bridging fibrovascular scar tissue that is highly aSMA+ at d14. There may also be tendon-specific differences as suggested by reviewers, since work from Dr. Dyment previously showed that the patellar tendon defect is bridged by paratenon cells without significant activity of the intrinsic tenocytes (in this model however, full transection was not carried out). Single cell RNASeq data from Dr. Chris Mendias’s group also show significant transcriptional differences between specific mouse tendons; these differences may result in different healing responses. Since this manuscript does not include any data from adult injury, we do not include extensive discussion of these points related to adult tendon injury models, but hopefully this response addresses reviewers’ questions.

6) Figure 3D: what do the authors attribute the massive variation in the # of TdTomato+ cell recruitment in inhibitor treated animals to? These data should also be analyzed as percentage of total cells in the bridging tissue. This will demonstrate whether inhibition affects all cell recruitment or specifically Scx^lin^.

To determine whether this variation is due to a single outlier or natural variability in healing, we acquired more ScxCreERT2/RosaT mice (n=2 Carrier and n=1 SB-431542). While the difference between treatment groups is now significant (p<0.05), the variability in the Carrier treated group remains. We therefore believe this is simply due to natural biological variation in the healing process. New graphs representing recruitment as percentage is also now included and we thank the reviewers for this suggestion. Interestingly, we find that total DAPI cells is decreased with SB-431542 treatment and Scx^lin^ cell recruitment is no longer different once normalized. However, no difference in total DAPI was observed in TBR2 group vs. WT and normalized Scx^lin^ recruitment is still different. This suggests to us that in the Tgfbr2 mutants, there is compensatory recruitment of non-Scx^lin^ cells when Scx^lin^ cells are not recruited. With SB-431542 treatment however, the compensation does not happen and total DAPI is reduced. We have now updated the Results subsection “,TGFβ signaling in neonatal tenocytes is required for cell recruitment after injury”, Discussion, and Figure 3.

7) Figure 4: In addition to normalization to carrier and WT for Edu labelling, the total % of proliferative cells would be informative about effects of Tgfb inhibition or deletion on general cellular dynamics.

The requested data are now included. The new data suggests that while Scx^lin^ cell proliferation is not affected with SB-431542 treatment, proliferation of non-Scx^lin^ cells may be decreased at d3 (p=0.07) while Tgfbr2^Scx^ mutants are not affected. Combined with our new results for DAPI quantification at d14, this suggests that the reduction in non-Scx^lin^ cells at d14 may stem in part from reduced proliferation at the earlier stage. Results subsection “TGFβ signaling is required for tenocyte migration but not proliferation” and Figure 4 have been updated.

8) Supplementary Figure 3/Figure 7: How was intensity of TGFb ligand staining quantified? Since I believe the authors main point is that there are spatial differences it would be appropriate to have these data on the same graph, and to have representative images from the gap space. In addition, what are the two different stains (purple/green) in 7A? Ideally, quantitative analyses of ligand levels via ELISA would be most convincing, but I recognize the technical challenges with this.

Staining intensity was quantified in ImageJ to obtain mean intensity in a region of interest. In tendons, the region of interest was chosen based on ScxGFP signal. In the gap space where there is no ScxGFP signal at d3, the region of interest was selected based on DAPI. The images have now been updated to show representative selected regions. As suggested by the reviewers, we now include data in the same graph and representative images from the gap space are now included (updated Figure 7). The green (ScxGFP) and purple (TGFβ1,2,3) signals are also now clearly labeled.

We also provide ELISA results for d14 for active TGFβ1 in Figure 7. We wish we were able to assay total TGFβ1 as well for this but did not have enough protein left. (See Results subsection “Increased TGFβ ligand production in injured tendon depends on TGFβ signaling”).

9) TGFb is known to stimulate proliferation of cells in culture. It appears that the cells were not serum starved just prior to the scratch assay or a proliferation inhibitor (e.g., Mitomycin C) was not added in one of the groups. Therefore, it's difficult to determine whether the changes were due to migration or proliferation. It would also be a nicer parallel to the in vivo studies to do the scratch assay with the Tgfbr2 cKO cells.

Cells were serum-starved prior to scratch assay. We apologize this detail was missing from the Materials and methods (now included). We have now repeated this experiment and included EdU quantification of proliferation during the scratch assay. Since we no longer have the Tgfbr2 floxed line, we are unable to carry out the experiment recommended. However, to distinguish tenocyte vs. non-tenocyte/epitenon migration/proliferation, we include ScxCreERT2 labeling in the assay and have quantified these cells. Note that Scx^lin^ cells were labeled in vivo at P2, P3 as normal and non-injured tendons harvested at P7 for the culture assay. The Materials and methods, Results subsection “TGFβ signaling is required for tenocyte migration but not proliferation”, and Figure 5 have been updated.

10) It is recommended not to call aSMA+ cells in this paper only myofibroblasts. Some of them are likely myofibroblasts but not all. Even though aSMA is a marker of a myofibroblast, all mesenchymal cells that express aSMA are not myofibroblasts. aSMA is expressed during expansion of progenitor cell populations that contribute to tissue healing. These cells go on to differentiate into ScxGFP+ cells, osteocytes, fibrocartilage, and odontoblasts. Myofibroblasts, on the other hand, are a more mature cell type. The key defining features of myofibroblasts are the de novo development of in vivo stress fibers and contractile forces. Stress fibers containing aSMA produce higher contractile forces, which is why myofibroblasts express aSMA. The fact that aSMA expression was not inhibited at this early stage by SB-431542, despite TGFb being a potent inducer of myofibroblasts (as stated in the third paragraph of the Discussion), also indicates that these cells may not be myofibroblasts.

This is a critical point and we agree completely. The language has been modified throughout the text, the discussion on myofibroblasts removed, and new discussion on aSMA^lin^ cells included (Discussion).

11) It is suggested that the authors add uninjured limbs to the main figure instead of in Figure 2—figure supplement 1. Panels A and B have the uninjured limbs so it would balance the figure nicely.

The uninjured limbs have been added to Figure 2.

12) Figure 4C – it would help to indicate on the graph that proliferation of the Scx^lin^ cells is reported.

The graph has been updated as recommended.

13) Figure 6—figure supplement 1 – Was aSMA staining higher at D14 in SB-431542-treated group? This could further support that healing is diminished/delayed in the treated group.

We returned to our images and quantified the staining using DAPI to select the region of interest. To our surprise, we do find that there is a difference in the SB-431542 treated group. We also quantified aSMA staining for Tgfbr2^Scx^ mutants at d14 and found no difference. This is now included in the text and updated Figure 6—figure supplement 1, we thank the reviewers for this suggestion (subsection “Non-Scx^lin^ tenogenic cells also contribute to neotendon formation”).

14) The titles of Figures 7 and 8 are the same.

The title for Figure 8 has been corrected, thank you (subsection “Non-Scx^lin^ tenogenic cells also contribute to neotendon formation”).

References:

Alyoussef, A. (2018). Blocking TGF-β type 1 receptor partially reversed skin tissue damage in experimentally induced atopic dermatitis in mice. Cytokine 106, 45-53.

Araujo-Jorge, T. C., Waghabi, M. C., Bailly, S. and Feige, J. J. (2012). The TGF-β pathway as an emerging target for Chagas disease therapy. Clin Pharmacol Ther 92, 613-621.

Bayomi, H. S., Elsherbiny, N. M., El-Gayar, A. M. and Al-Gayyar, M. M. (2013). Evaluation of renal protective effects of inhibiting TGF-β type I receptor in a cisplatin-induced nephrotoxicity model. Eur Cytokine Netw 24, 139-147.

Chowdhury, B. P., Das, S., Majumder, S., Halder, K., Ghosh, S., Biswas, S., Bandyopadhyay, S. and Majumdar, S. (2015). Immunomodulation of host-protective immune response by regulating Foxp3 expression and Treg function in Leishmania-infected BALB/c mice: critical role of IRF1. Pathog Dis 73, ftv063.

Koh, R. Y., Lim, C. L., Uhal, B. D., Abdullah, M., Vidyadaran, S., Ho, C. C. and Seow, H. F. (2015). Inhibition of transforming growth factor-β via the activin receptor-like kinase-5 inhibitor attenuates pulmonary fibrosis. Mol Med Rep 11, 3808-3813.

Lu, L., Bai, X., Cao, Y., Luo, H., Yang, X., Kang, L., Shi, M. J., Fan, W. and Zhao, B. Q. (2018). Growth Differentiation Factor 11 Promotes Neurovascular Recovery After Stroke in Mice. Front Cell Neurosci 12, 205.

Miao, Z. F., Zhao, T. T., Wang, Z. N., Miao, F., Xu, Y. Y., Mao, X. Y., Gao, J., Wu, J. H., Liu, X. Y., You, Y., et al. (2014). Transforming growth factor-beta1 signaling blockade attenuates gastric cancer cell-induced peritoneal mesothelial cell fibrosis and alleviates peritoneal dissemination both in vitro and in vivo. Tumour Biol 35, 3575-3583.

Mordasky Markell, L., Perez-Lorenzo, R., Masiuk, K. E., Kennett, M. J. and Glick, A. B. (2010). Use of a TGFbeta type I receptor inhibitor in mouse skin carcinogenesis reveals a dual role for TGFbeta signaling in tumor promotion and progression. Carcinogenesis 31, 2127-2135.

Schindeler, A., Morse, A., Peacock, L., Mikulec, K., Yu, N. Y., Liu, R., Kijumnuayporn, S., McDonald, M. M., Baldock, P. A., Ruys, A. J., et al. (2010). Rapid cell culture and pre-clinical screening of a transforming growth factor-β (TGF-β) inhibitor for orthopaedics. BMC Musculoskelet Disord 11, 105.

Tanaka, H., Shinto, O., Yashiro, M., Yamazoe, S., Iwauchi, T., Muguruma, K., Kubo, N., Ohira, M. and Hirakawa, K. (2010). Transforming growth factor β signaling inhibitor, SB-431542, induces maturation of dendritic cells and enhances anti-tumor activity. Oncol Rep 24, 1637-1643.

Waghabi, M. C., Keramidas, M., Calvet, C. M., Meuser, M., de Nazare, C. S. M., Mendonca-Lima, L., Araujo-Jorge, T. C., Feige, J. J. and Bailly, S. (2007). SB-431542, a transforming growth factor β inhibitor, impairs Trypanosoma cruzi infection in cardiomyocytes and parasite cycle completion. Antimicrob Agents Chemother 51, 2905-2910.

Wei, W., Ma, B., Li, H. Y., Jia, Y., Lv, K., Wang, G., Zhang, J., Zhu, S., Tang, H., Sheng, Z., et al. (2010). Biphasic effects of selective inhibition of transforming growth factor beta1 activin receptor-like kinase on LPS-induced lung injury. Shock 33, 218-224.

Zhang, X. Y., Liu, Z. M., Zhang, H. F., Li, Y. S., Wen, S. H., Shen, J. T., Huang, W. Q. and Liu, K. X. (2016). TGF-beta1 improves mucosal IgA dysfunction and dysbiosis following intestinal ischaemia-reperfusion in mice. Journal of cellular and molecular medicine 20, 1014-1023.

Zhou, H. Q., Liu, M. S., Deng, T. B., Xie, P. B., Wang, W., Shao, T., Wu, Y. and Zhang, P. (2019). The TGF-β/Smad Pathway Inhibitor SB431542 Enhances The Antitumor Effect Of Radiofrequency Ablation On Bladder Cancer Cells. Onco Targets Ther 12, 7809-7821.

Zi, Z., Chapnick, D. A. and Liu, X. (2012). Dynamics of TGF-β/Smad signaling. FEBS letters 586, 1921-1928.